# Cell type composition and circuit organization of clonally related excitatory neurons in the juvenile mouse neocortex

Cathryn R Cadwell[1,2,3†]\*, Federico Scala[1,2], Paul G Fahey[1,2], Dmitry Kobak[4], Shalaka Mulherkar[1], Fabian H Sinz[1,2,5,6], Stelios Papadopoulos[1,2], Zheng H Tan[1,2], Per Johnsson[7], Leonard Hartmanis[7], Shuang Li[1,2], Ronald J Cotton[1,2], Kimberley F Tolias[1,8], Rickard Sandberg[7], Philipp Berens[4,5], Xiaolong Jiang[1,2,9]\*, Andreas Savas Tolias[1,2,10]\*

[1]Department of Neuroscience, Baylor College of Medicine, Houston, United States; [2]Center for Neuroscience and Artificial Intelligence, Baylor College of Medicine, Houston, United States; [3]Department of Anatomic Pathology, University of California San Francisco, San Francisco, United States; [4]Institute for Ophthalmic Research, University of Tübingen, Tübingen, Germany; [5]Department of Computer Science, University of Tübingen, Tübingen, Germany; [6]Interfaculty Institute for Biomedical Informatics, University of Tübingen, Tübingen, Germany; [7]Department of Cell and Molecular Biology, Karolinska Institutet, Stockholm, Sweden; [8]Verna and Marrs McLean Department of Biochemistry and Molecular Biology, Baylor College of Medicine, Houston, United States; [9]Jan and Dan Duncan Neurological Research Institute at Texas Children's Hospital, Houston, United States; [10]Department of Electrical and Computer Engineering, Rice University, Houston, United States

\*For correspondence:
Cathryn.Cadwell@ucsf.edu (CRC);
xiaolonj@bcm.edu (XJ);
astolias@bcm.edu (AST)

Present address: [†]Department of Pathology, University of California San Francisco, San Francisco, CA, United States

Competing interests: The authors declare that no competing interests exist.

**Abstract** Clones of excitatory neurons derived from a common progenitor have been proposed to serve as elementary information processing modules in the neocortex. To characterize the cell types and circuit diagram of clonally related excitatory neurons, we performed multi-cell patch clamp recordings and Patch-seq on neurons derived from *Nestin*-positive progenitors labeled by tamoxifen induction at embryonic day 10.5. The resulting clones are derived from two radial glia on average, span cortical layers 2–6, and are composed of a random sampling of transcriptomic cell types. We find an interaction between shared lineage and connection type: related neurons are more likely to be connected vertically across cortical layers, but not laterally within the same layer. These findings challenge the view that related neurons show uniformly increased connectivity and suggest that integration of vertical *intra*-clonal input with lateral *inter*-clonal input may represent a developmentally programmed connectivity motif supporting the emergence of functional circuits.

## Introduction

The mammalian neocortex carries out complex mental processes such as cognition and perception through the interaction of billions of neurons connected by trillions of synapses. We are just beginning to understand how networks of neurons become wired together during development to give rise to cortical computations (*Polleux et al., 2007*; *Cadwell et al., 2019*). During cortical neurogenesis, which lasts from approximately embryonic day 10.5 (E10.5) through E17.5 in the mouse (*Caviness et al., 1995*; *Takahashi et al., 1996*), radial glial cells undergo asymmetric division to generate postmitotic excitatory neurons that migrate radially to populate the cortical plate. Neurogenesis occurs in an inside-out gradient, such that early born neurons occupy the deep cortical layers and

later-born neurons reside in progressively more superficial layers (*Angevine and Sidman, 1961*; *Rakic, 1974*; *Caviness et al., 1995*). The ventricular zone is subdivided by glial septa into well-defined columns of precursor or stem cells referred to as 'proliferative units' that give rise to a radial unit of clonally related excitatory neurons, sometimes referred to as an *ontogenetic column* (*Torii et al., 2009*; *Kriegstein and Noctor, 2004*; *Noctor et al., 2001*; *Noctor et al., 2007*; *Rakic, 1988*). However, these radial units of clonally related neurons are only loosely clustered and are heavily intermixed with numerous nearby unrelated neurons (*Walsh and Cepko, 1988*; *Tan et al., 1995*) and there is substantial tangential migration of clonally related neurons as they traverse the subventricular zone and intermediate zone *en route* to the developing cortical plate (*Torii et al., 2009*). In contrast to excitatory neurons, inhibitory interneurons are generated in the ganglionic eminences and migrate tangentially to disperse throughout the developing cortical mantle (*Letinic et al., 2002*; *Kriegstein and Noctor, 2004*; *Tan et al., 1998*; *Mayer et al., 2015*).

Recent advances in single-cell RNA-sequencing technology (*Tang et al., 2009*; *Picelli et al., 2013*; *Picelli et al., 2014a*) have enabled unbiased cell type classification in heterogeneous tissues including the cerebral cortex (*Zeisel et al., 2015*; *Tasic et al., 2016*; *Tasic et al., 2018*). In contrast to inhibitory interneurons, excitatory neurons in the adult mouse (*Tasic et al., 2018*) and developing human (*Nowakowski et al., 2017*) cortex are largely region-specific at the level of transcriptomic cell types, with several dozens of excitatory cell types per area (*Tasic et al., 2018*; *Hodge et al., 2019*). While it is well-established that the vast majority of cells within radial clones are excitatory neurons (*Tan et al., 1998*), it remains controversial whether individual progenitors give rise to the full diversity of excitatory neuron cell types within a given cortical area, or only to a restricted subset of transcriptomic cell types (*Franco et al., 2012*; *Gil-Sanz et al., 2015*; *Eckler et al., 2015*; *Kaplan et al., 2017*; *Llorca et al., 2019*).

A series of studies using a retroviral lineage tracing method has suggested that clonally related excitatory neurons are more likely to be synaptically connected to each other (*Yu et al., 2009*; *Yu et al., 2012*; *He et al., 2015*) and have similar preferred orientations in primary visual cortex (V1) compared to unrelated neurons (*Li et al., 2012*), providing support for the long-standing hypothesis that radial clones may constitute elementary circuit modules for information processing in the cortex (*Rakic, 1988*; *Mountcastle, 1997*; *Buxhoeveden and Casanova, 2002*). The projection pattern of vertical, across-layer connections between related neurons was qualitatively similar to the canonical circuit of layer-specific connections in adult cortex (*Yu et al., 2009*); however, a direct comparison of related and unrelated pairs for each layer-specific connection type was not done, and lateral connections between clonally related cells within the same cortical layer were not examined. Therefore, it remains unclear whether all local connections are uniformly increased between clonally related excitatory neurons, although this assumption has become dogma in the field (*Li et al., 2018*). Given the complexity of the local cortical circuit and the different functional roles of layer-defined connections (*Lefort et al., 2009*; *Feldmeyer, 2012*; *Lübke et al., 2000*; *Lübke et al., 2003*), clarifying the effect of cell lineage on the underlying layer-specific connectivity matrix may have important implications regarding the mechanism and functional purpose of lineage-associated connectivity. The difficulty of multi-patching experiments combined with the relatively low connectivity rates between excitatory neurons (*Jiang et al., 2015*; *Markram et al., 1997*; *Barth et al., 2016*; *Jiang et al., 2016*) necessitate testing a very large number of connections and pose an enormous technical challenge to fully answer this question.

Using an enhancer trap Cre line to label neural progenitors at an earlier developmental stage, yielding much larger clones (approximately 670–800 neurons per clone compared to 4–6 neurons per clone in *Yu et al., 2009*), a separate group has also reported an association between shared lineage and orientation tuning in V1 (*Ohtsuki et al., 2012*), although with a much smaller effect size than reported in *Li et al. (2012)* for small clones. Another study examined lateral connections within layer 4 (L4) of large clones labeled in chimeric mice at approximately E3.5 (*Tarusawa et al., 2016*) and found transient increases in connectivity between clonally related cells. These studies raise the possibility that shared lineage may serve as an important predictor of large-scale functional circuits even in more distantly related neurons derived from symmetrically dividing neural stem cells (*Smith and Fitzpatrick, 2012*). However, the relationship between lineage and the layer-specific connectivity matrix has not been systematically studied in large clones, and there remains a major disconnect in scale between studies of lineage-associated connectivity and the development of larger, functional cortical units that are thought to implement computations.

Here we use a tamoxifen-inducible Cre-lox system to label progenitors at approximately the onset of neurogenesis, resulting in intermediate-sized translaminar clones that span cortical layers 2–6 and ask: (a) what is the cell type composition of individual translaminar clones and (b) does the layer-specific connectivity matrix among clonally related excitatory neurons differ from unrelated neurons. We find that clones of excitatory neurons labeled at embryonic day 10.5 (E10.5) are derived from two radial glia on average, and are composed of diverse transcriptomic cell types. Vertical connections linking cells across cortical layers, and in particular connections from upper cortical layers onto layer 5 (L5) excitatory neurons, are selectively increased among clonally related neurons compared to unrelated neurons. In contrast, clonally related excitatory neurons are not preferentially connected laterally, within the same cortical layer, despite being located in close proximity to one another. These findings argue against the prevailing view that preferential connectivity between clonally related excitatory neurons mirrors that of the canonical cortical circuit, and suggest that integration of vertical input from related neurons with lateral inputs from unrelated neurons may represent a developmentally programmed motif for assembling functional cortical circuits.

## Results

### Tamoxifen induction at E10.5 generates translaminar clones spanning cortical layers 2–6

To label radial clones, we induced sparse recombination in progenitors at approximately the onset of neurogenesis using a tamoxifen-inducible Cre-lox transgenic system driven by the *Nes* promoter (*Figure 1A,B*). In contrast to viral lineage tracing methods, which are routinely performed at E12.5 or later (*Yu et al., 2009*; *Yu et al., 2012*; *Li et al., 2012*), and enhancer trap methods (*Ohtsuki et al., 2012*) that do not precisely control the timing and density of labeling, our tamoxifen-inducible system can reproducibly and sparsely label progenitor cells in the developing forebrain across a range of timepoints (*Figure 1C*). We first compared clone size and layer distribution at postnatal day 10 (P10) following tamoxifen administration at E9.5, E10.5, and E11.5. Tamoxifen administration at E9.5 resulted in clones containing on average 196 neurons (median, interquartile range [IQR] 75–346 neurons) and spanning 347 µm in width (median, IQR 223–402 µm; *Figure 1D,E*). Tamoxifen administration at E10.5 resulted in clones containing on average 60 neurons (median, IQR 26–117) and spanning 224 µm in width (median, IQR 140–298 µm; *Figure 1D,E*). Tamoxifen administration at E11.5 resulted in clones containing on average 39 neurons (median, IQR 24–63) and spanning 220 µm in width (median, IQR 162–276 µm; *Figure 1D,E*), similar to the width of clones labeled at E10.5. However, a substantial fraction of clones labeled at E11.5 did not contain neurons in the deep cortical layers 5–6 (10/50 clones, 20%), which was only rarely seen in clones labeled by induction at E10.5 (1/39 clones, 3%), and never following induction at E9.5 (0/35 clones, 0%; (*Figure 1F* and *Figure 1—figure supplement 1*). These data suggest that progenitors labeled at E11.5 or later may be diverse, with a subset of these no longer generating deep layer neurons. Given the variability in clones seen with induction at E11.5, we focused our study on clones generated by tamoxifen induction at E10.5.

Given that Nestin is expressed by neuroepithelial stem cells and ventricular radial glia (*Hockfield and McKay, 1985*; *Misson et al., 1988*) and that the transition between neuroepithelial stem cells and radial glia occurs between E10.25 and E11.5 (*Misson et al., 1988*; *Anthony et al., 2004*; *Nowakowski et al., 2011*), it is possible that a subset of our clones are derived from neuroepithelial stem cells. However, there is a substantial delay in onset and prolonged pharmacokinetic activity of tamoxifen following oral administration (*Robinson et al., 1991*), and an inherent imprecision in the timing of pregnancies. Therefore, to empirically determine the type of progenitors labeled in our experimental model, embryos were sacrificed at E12.5 following tamoxifen administration at E10.5 and immunohistochemistry for Pax6 (expressed in radial glial and intermediate progenitors, *Götz et al., 1998*) and Tbr2 (also known as Eomes, a marker of intermediate progenitors, *Englund et al., 2005*; *Mihalas et al., 2016*) were performed to characterize the type and number of progenitors within individual clones (*Figure 1G,H* and *Figure 1—figure supplement 2*). On average, individual clones contained two radial glia (median, range 1–4, *Figure 1I* and *Figure 1—figure supplement 2*). As described by *Rakic (1988)*, proliferative units can be identified as well-defined columns of precursor or stem cells within the ventricular zone; it has been proposed that the polyclonal

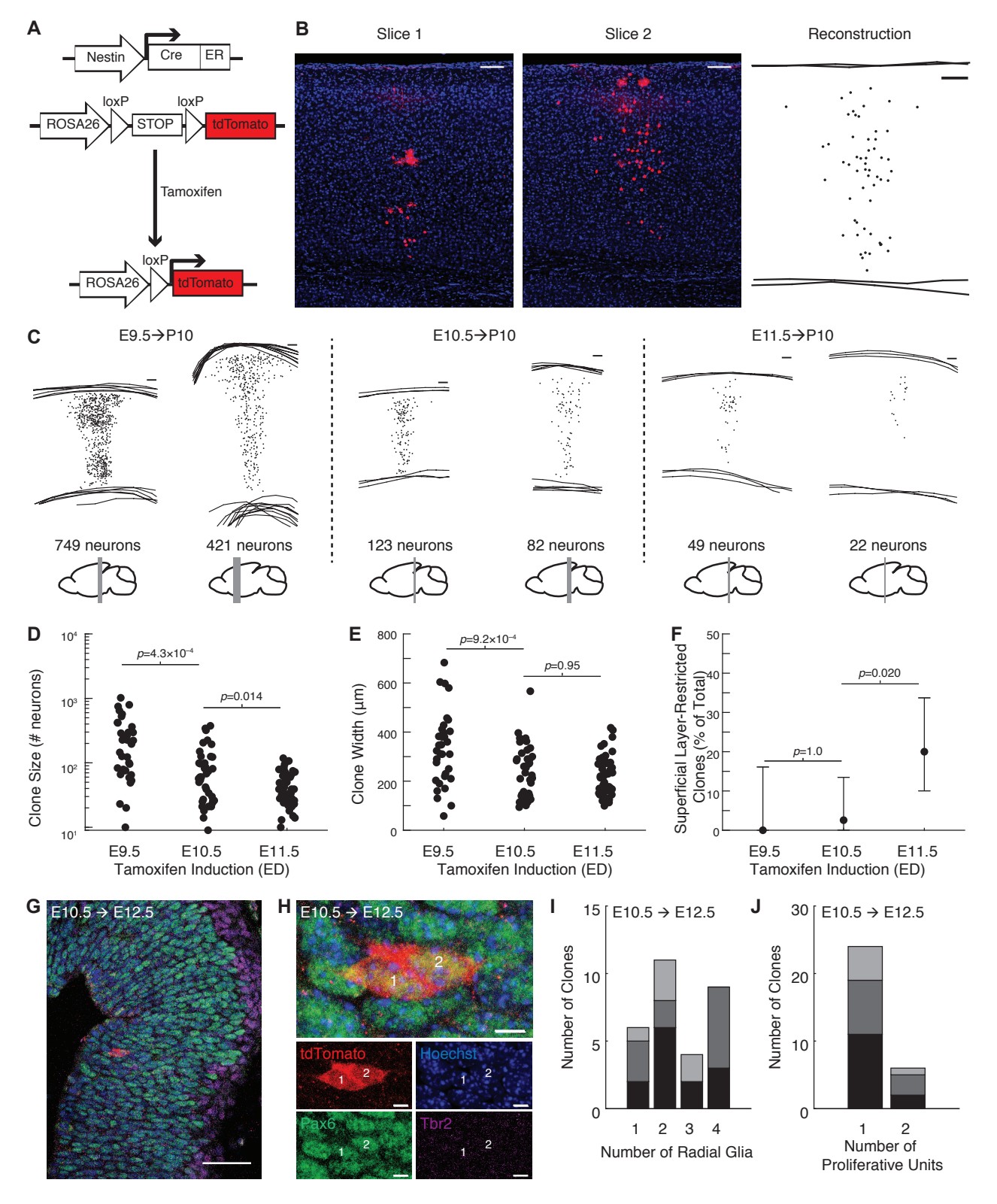

**Figure 1.** Tamoxifen induction at E10.5 generates translaminar clones spanning cortical layers 2–6. (A) Schematic of tamoxifen-inducible Cre-loxP system for lineage tracing. (B) Manual reconstruction of a clone across multiple slices. In this example, larger red spots are morphologically consistent with glial cells at high magnification. Scale bar: 100 µm. (C) Examples of reconstructed clones labeled at E9.5, E10.5 or E11.5. Scale bars: 100 µm. Differences in thickness are due to tangential sectioning of the cortex rostrally (approximate rostrocaudal position of each clone is shown in gray

*Figure 1 continued on next page*

*Figure 1 continued*

beneath the reconstruction). (**D and E**) Number of neurons (**D**) and clone width (**E**) at postnatal day 10 following tamoxifen induction at E9.5, E10.5, or E11.5 (n = 35, 39, and 50 clones; n = 3, 4 and 3 mice per condition; *p*-values computed using Wilcoxon rank sum). (**F**) Percent of clones that are do not contain neurons in L5 or L6 following tamoxifen induction at E9.5, E10.5, or E11.5 (n = 0/35, 1/39, and 10/50 clones; n = 3, 4 and 3 mice per condition; *p*-values computed using Fisher's exact test). Error bars show 95% Clopper-Pearson confidence intervals. (**G**) Developing cortical plate with immunohistochemical staining for Pax6 (green), Tbr2 (magenta), and tdTomato (red) to characterize progenitors within individual clones at E12.5 following tamoxifen induction at E10.5. Scale bar, 50 µm. (**H**) High magnification of clone shown in G, demonstrating two PAX6+/Tbr2- radial glia within one proliferative unit. Scale bar, 5 µm. In panels (**G**) and (**H**), the images are oriented with the ventricular zone on the left and the pial surface on the right. (**I and J**) Summary of the number of radial glia and the number of proliferative units per clone for all clones analyzed at E12.5 (n = 30 clones from three mice). Shades of gray in panels (**I** and **J**) denote data obtained from different animals for each treatment condition. See also *Figure 1—figure supplements 1* and *2* and *Figure 1—source datas 1* and *2*.

The online version of this article includes the following source data and figure supplement(s) for figure 1:

**Source data 1.** Clone quantification data, related to *Figure 1*.
**Source data 2.** Number and type of progenitors per clone, related to *Figure 1*.
**Figure supplement 1.** Additional examples and quantitative analyses of clones induced at E11.5 that are missing deep cortical layers, expanding on *Figure 1C,F*.
**Figure supplement 2.** Progenitor composition of clones examined at E12.5 following tamoxifen administration at E10.5, additional examples expanding on *Figure 1G–J*.

precursors within a proliferative unit collectively give rise to a population of postmitotic neurons that share a common radial glial scaffold and form a morphologically identifiable stack of neurons in the cortex termed an 'ontogenetic' or 'embryonic' column (*Rakic, 1988*). Although no well-accepted criteria are available for defining the boundaries of a proliferative unit, we found that the vast majority of our clones labeled at E10.5 were arranged in a single vertical track at E12.5 consistent with their belonging to a single proliferative unit (24 out of 30 clones, 80%, *Figure 1J* and *Figure 1—figure supplement 2*). These findings show that tamoxifen administration at E10.5 labels predominantly individual radial glia and neuroepithelial stem cells in their final 1–2 cycles of symmetric cell division.

## Region-specific differences in gene expression are present in juvenile mouse neocortex

Recent transcriptomic cell atlases of adult mouse neocortex showed that different cortical areas are composed of distinct transcriptomic excitatory neuron cell types (*Tasic et al., 2018*; *Saunders et al., 2018*). Given that our lineage tracing strategy labels translaminar clones across many cortical regions, we next asked whether region-specific excitatory neurons are present also in juvenile mouse neocortex.

To test this, we cut acute parasagittal slices spanning primary visual (V1) and primary somatosensory (SS1) cortices from juvenile (P15-P20) mice. Translaminar clones were identified by their intrinsic fluorescence (*Figure 2A,B*), and the contents of individual labeled neurons as well as nearby unlabeled control neurons were aspirated through a patch pipette following brief electrophysiological recording using our recently described Patch-seq protocol (*Cadwell et al., 2017*; *Cadwell et al., 2016*; *Scala et al., 2019*). We analyzed 206 neurons (after quality control, see *Figure 2—figure supplement 1* and Methods) which had approximately 0.36 million uniquely mapping reads (median; IQR 0.17–0.69 million reads; *Figure 2C*) and approximately 7000 genes detected (median: 7007; IQR 6152–7920 genes; *Figure 2D*) on average per cell. For all downstream analyses, we used 12,841 genes that had on average >1 count/cell (see Methods). Alternatively filtering genes based on the number of cells expressing each gene had no substantial effect on our results (data not shown). There were modest differences in the average library size (*Figure 2—figure supplement 1E,F*) and number of genes expressed (*Figure 2—figure supplement 1I,J*) between different cortical areas and different layers. However, there were no significant differences between tdTomato-positive (clonally related) and tdTomato-negative (nearby unrelated) neurons in either of these two measures (*Figure 2—figure supplement 1G,K*). Count data were normalized using a pool-based strategy developed specifically for single-cell RNA sequencing analysis (*Lun et al., 2016*). Size factors largely correlated with library size (*Figure 2—figure supplement 1H*), suggesting that our cell population is relatively homogenous and that systematic differences in gene counts in our dataset are driven

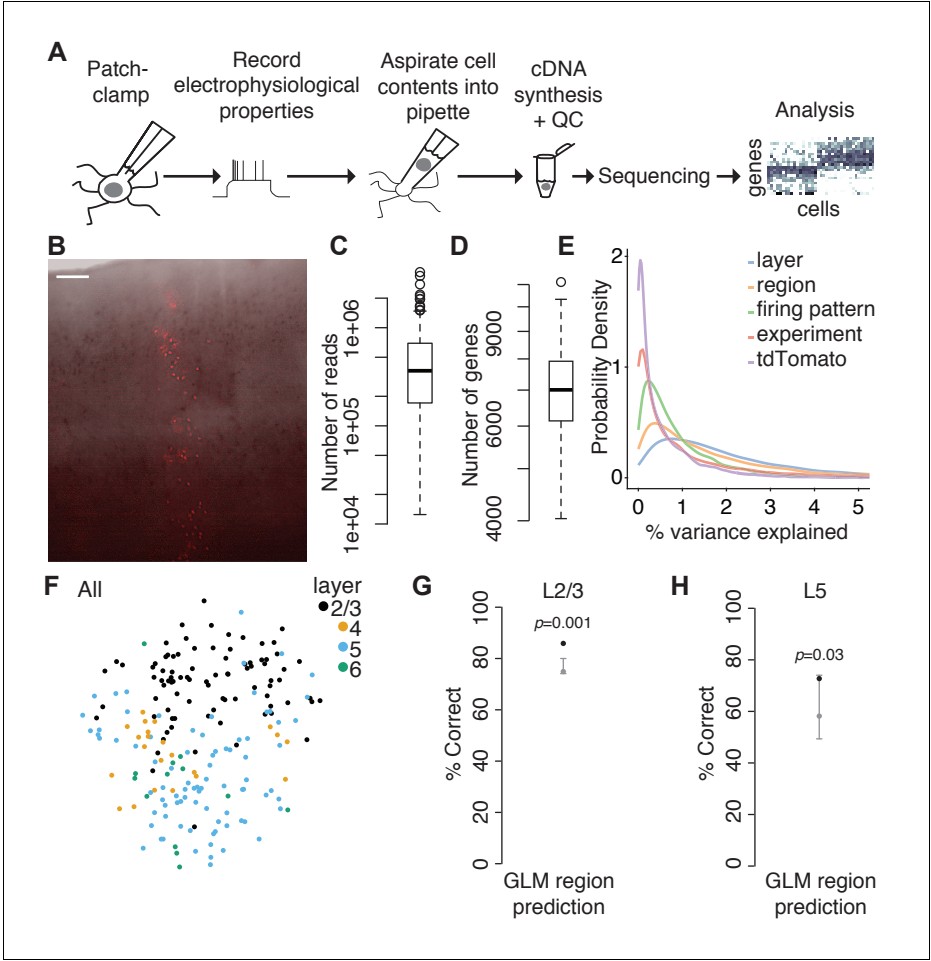

**Figure 2.** Region-specific differences in gene expression are present in juvenile mouse neocortex. (**A**) Overview of experimental approach using Patch-seq. (**B**) Example tdTomato-positive translaminar clone spanning cortical layers 2–6 in an acute cortical slice used for Patch-seq experiments. Overlay of bright field and fluorescence image was performed in Adobe Photoshop. Scale bar: 100 µm. (**C and D**) Box plots showing library size (**C**) and number of genes detected (**D**) for all cells passing quality control criteria (n = 206). (**E**) Density plot of the percent of variance in normalized log-expression values explained by different experimental factors. Each curve corresponds to the variance in gene expression across all genes (n = 12,841 genes) that can be explained by a single variable, with right-shifted curves reflecting variables that explain a higher fraction of the variance. (**F**) T-distributed stochastic neighbor embedding (t-SNE) plots using the top highly variable and correlated genes across all cells (n = 91 genes; n = 87, 22, 84, and 13 cells in layers 2/3, 4, 5, and 6, respectively), colored by layer position. (**G**) Performance of a generalized linear model (GLM) trained to predict region from gene expression data of L2/3 neurons (n = 12,841 genes and 85 cells) with model performance (black dot) compared to the chance-level performance estimated using shuffled data (gray, mean and 95% coverage interval; one-tailed *p*-value computed from shuffled data, shuffling region). (**H**) Performance of a GLM trained to predict region from gene expression data of L5 neurons (n = 12,841 genes and 77 cells) as described in (**G**). See also *Figure 2—figure supplements 1* and *2* and *Figure 2—source data 1*.

The online version of this article includes the following source data and figure supplement(s) for figure 2:

**Source data 1.** Gene expression data, related to *Figure 2*.

**Figure supplement 1.** Quality control criteria for single-cell RNA-sequencing data, related to *Figure 2C,D*.

**Figure supplement 2.** Expression of top highly variable genes, related to *Figure 2F–H*.

primarily by technical factors such as capture efficiency and sequencing depth. The normalized counts (*Figure 2—source data 1*) were used in all subsequent analyses.

Consistent with recent studies (*Zeisel et al., 2015*; *Tasic et al., 2016*; *Tasic et al., 2018*; *Nowakowski et al., 2017*), we found that layer position and cortical region were the strongest

predictors of transcriptomic variability among excitatory neurons in our dataset (*Figure 2E* and *Figure 2—figure supplement 2*). Dimensionality reduction using t-distributed stochastic neighbor embedding (t-SNE) revealed that neurons clustered primarily by layer position (*Figure 2F*), and a cross-validated generalized linear model (GLM) could predict layer position from the gene expression data with approximately 80% accuracy (data not shown). Within L2/3 and L5, neurons from V1 and SS1 formed overlapping clusters in high-dimensional gene space (data not shown), but a cross-validated generalized linear model (GLM) could predict regional position slightly better than chance (*Figure 2G,H*), suggesting that L2/3 and L5 excitatory neurons in juvenile mouse neocortex have already begun differentiating along region-specific transcriptomic pathways. Thus, we reasoned that in order to determine whether individual progenitors are fated to generate a restricted subset of transcriptomic cell types, it would be critical to compare the cell type composition of clonally related neurons to nearby unlabeled neurons with a similar regional position.

## Translaminar clones labeled at E10.5 are composed of diverse transcriptomic subtypes of excitatory neurons

While most evidence supports a deterministic model of excitatory neurogenesis, whereby individual progenitors give rise to many different excitatory neuron cell types through progressive fate restriction (*Tan and Breen, 1993*; *Guo et al., 2013*; *Gao et al., 2014*), some studies suggest that a subset of progenitors are fate-restricted early on to give rise to layer-restricted excitatory neurons (*Franco and Müller, 2013*; *Franco et al., 2012*; *Gil-Sanz et al., 2015*). In addition, one recent study suggests that a large fraction of clones spanning the full thickness of the cortex are composed exclusively of corticocortical projection neurons (*Llorca et al., 2019*).

To characterize the diversity of cell types within our translaminar clones, we mapped our single-cell transcriptional profiles to a recently published cell type atlas of adult mouse cortex (*Tasic et al., 2018*). Most neurons patched in L2/3 mapped to L2/3 reference types (65/87, 74.7%), and similarly for L4 (15/22, 68.2%), L5 (54/84, 64.3%), and L6 (11/13, 84.6%) (*Figure 3A* and *Figure 3—source data 1*). The discrepancies were mostly due to some neurons mapping to a transcriptomic type from a neighboring layer: neurons from L2/3 mapping to L4 types (12/87, 13.8%), neurons from L5 mapping to L4 types (8/84, 9.5%), and neurons from L5 mapping to L6 types (19/84, 22.6%). These results are not surprising given that some transcriptomic cell types are also present in adjacent cortical layers (*Tasic et al., 2018*). Four cells mapped to interneuron types (two tdTomato-positive and two tdTomato-negative) and one cell mapped to an oligodendrocyte type (tdTomato-positive), which may reflect imperfect mapping of our dataset to the reference dataset or incidental aspiration of an adjacent cell. Area SS1 was not specifically profiled in the reference cell atlas; we found that cells from both V1 and SS1 in our dataset mapped predominantly to V1 excitatory neuron types (94.9% of V1 cells and 92.4% of SS1 cells; *Figure 3—source data 1*) with only a handful mapping to ALM excitatory neuron types (3.4% of V1 cells and 3.8% of SS1 cells). Of note, the quality of the mapping was equally good for V1 and SS1 cells (correlations to best matching cluster for SS1 cells, 0.78 ± 0.08; for V1 cells, 0.78 ± 0.07; mean ± SD; p=0.76, two-sample t-test), suggesting that the adult V1/ALM cell type atlas is an equally reasonable reference for excitatory neurons in juvenile V1 and SS1.

Collectively, the labeled neurons within our clones mapped to all of the broad excitatory cell classes (i.e. intratelencephalic [IT], pyramidal tract [PT], near-projecting [NP], and corticothalamic [CT]) in proportions similar to unlabeled control neurons (p=0.38, Chi-squared test; *Figure 3—figure supplement 1A,B*). We had at least three cells mapped to each of thirteen transcriptomic cell types defined in *Tasic et al. (2018)* (*Figure 3—source data 1*). There was a difference in the overall distribution of labeled and unlabeled neurons among these thirteen cell types (p=0.039, Chi-squared test; *Figure 3B*), and post-hoc comparisons revealed that the L2/3 IT VISp Rrad transcriptomic cell type was underrepresented in our labeled clones compared to unlabeled control neurons (p=0.023, Chi-squared test with Bonferroni correction for 13 comparisons, *Figure 3B*; p>0.05 for the other 12 transcriptomic cell types). These findings demonstrate that the progenitors labeled in our study collectively give rise to the majority of transcriptomic cell types present in the areas of cortex examined, but suggest that the L2/3 IT VISp Rrad transcriptomic cell type may arise from a distinct progenitor pool.

Within individual clones, we found that pairs of related neurons were no more likely to map to the same broad transcriptomic class (p=0.71, Chi-squared test; *Figure 3—figure supplement 1C*

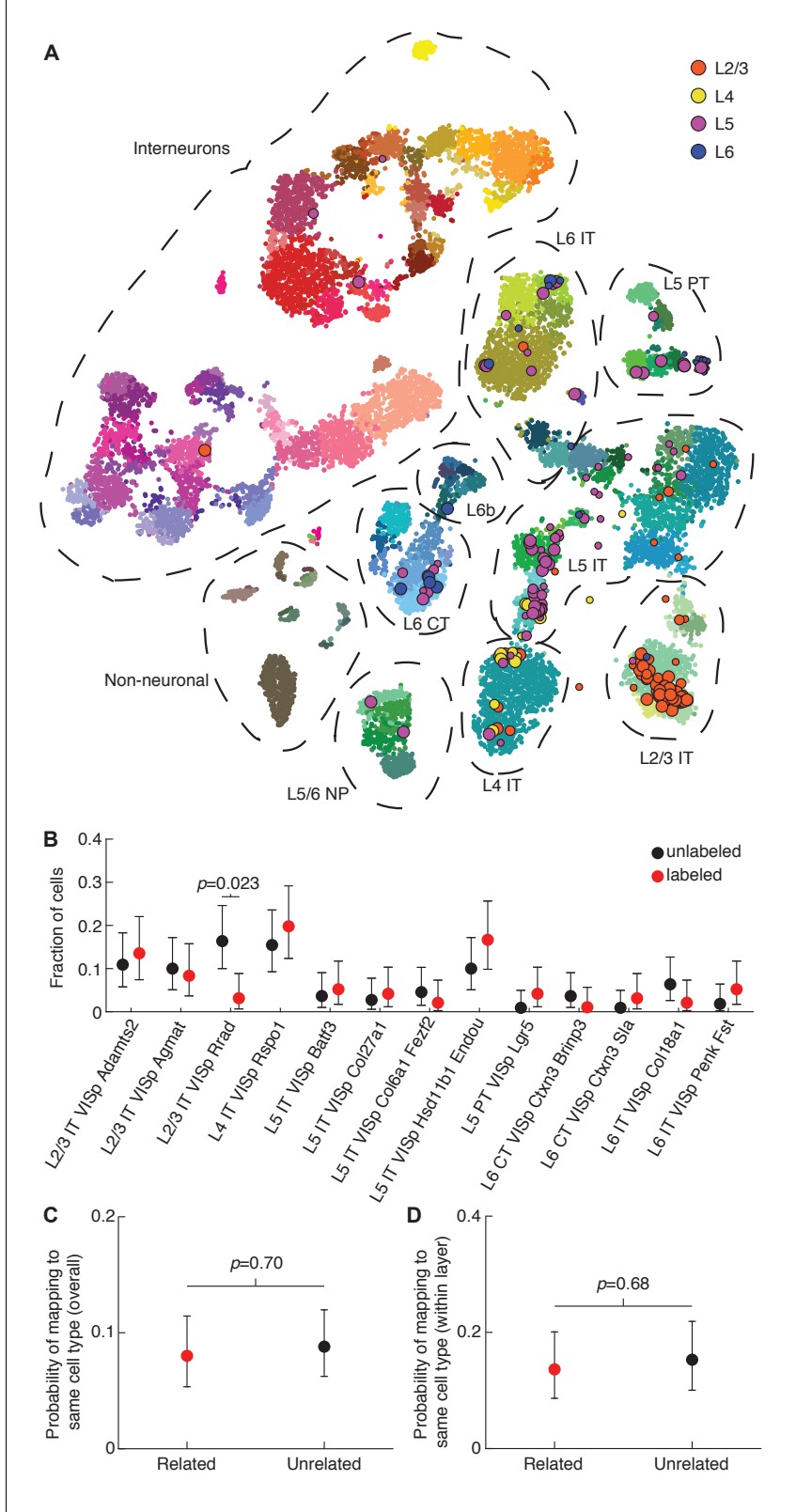

**Figure 3.** Translaminar clones labeled at E10.5 are composed of diverse transcriptomic subtypes of excitatory neurons. (A) T-distributed stochastic neighbor embedding (t-SNE) plot showing alignment of our Patch-seq data (data points with black outline, n = 87, 22, 84, and 13 cells in layers 2/3, 4, 5, and 6 respectively) with a recently published mouse cell type atlas (data points with no outline; n = 23,822; from *Tasic et al. (2018)*; colors denote

*Figure 3 continued*

transcriptomic types and are taken from the original publication). The t-SNE of the reference dataset and the positioning of Patch-seq cells were performed as described in *Kobak and Berens (2019)*, see Methods. The size of the Patch-seq data points denotes the precision of the mapping (see Materials and methods): small points indicate high uncertainty. (B) Fraction of labeled (n = 96) and unlabeled (n = 110) cells mapping to specific transcriptomic cell types (cell types with less than three neurons mapped are not shown; overall p=0.039, Chi-squared test). (C and D) Probability of related and unrelated cell pairs mapping to the same transcriptomic cell type either overall (C; n = 337 related pairs, n = 409 unrelated pairs, p=0.70, Chi-squared test) or conditioned on layer position (D; n = 154 related pairs, n = 157 unrelated pairs, p=0.68, Chi-squared test). For (B–D), error bars are 95% Clopper-Pearson confidence intervals and *p*-values are computed using Chi-squared test (in B, we used Bonferroni correction for each of the 13 post-hoc comparisons). See also *Figure 3—figure supplements 1* and *2* and *Figure 3—source data 1*.

The online version of this article includes the following source data and figure supplement(s) for figure 3:

**Source data 1.** Mapping to transcriptomic cell types, related to *Figure 3*.
**Figure supplement 1.** Translaminar clones labeled at E10.5 are composed of diverse classes of excitatory neurons, related to *Figure 3*.
**Figure supplement 2.** Transcriptomic diversity of individual translaminar clones, related to *Figure 3*.

and *Figure 3—figure supplement 2*), or specific cell type (p=0.70, Chi-squared test; *Figure 3C* and *Figure 3—figure supplement 2*) compared to pairs of nearby unrelated neurons from the same cortical region. One possibility is that our clones appeared to contain different cell types simply because they spanned multiple layers, but within a layer they may still be restricted to give rise to a particular cell type. To test this, we also compared pairs of clonally related neurons within the same cortical layer to pairs of unrelated neurons in the same layer and found no detectable bias among clonally related neurons to belong to the same broad (p=0.76, Chi-squared test; *Figure 3—figure supplement 1D*) or specific (p=0.68, Chi-squared test; *Figure 3D*) transcriptomic cell type. While these findings cannot exclude the possibility of fate restriction of individual radial glia or intermediate progenitors labeled at later stages, they suggest that individual progenitors labeled at E10.5 give rise to a broad range of excitatory cell types.

## Vertical, across-layer connections are selectively increased between excitatory neurons in translaminar clones

To determine whether clonally related neurons within our translaminar clones were preferentially connected, we performed a separate series of experiments using multiple simultaneous whole-cell recordings as previously described (*Jiang et al., 2015*), targeting up to eight neurons simultaneously including both clonally related cells and nearby unlabeled control cells (*Figure 4A,B*). In total, we patched 592 neurons (310 labeled and 282 unlabeled) from 86 clones in 43 mice. The layer position of each cell was determined using differential interference contrast imaging at the time of recording and later confirmed using avidin-biotin-peroxidase staining as previously described (*Jiang et al., 2015*; *Cadwell et al., 2017*; *Scala et al., 2019*). The cells were distributed throughout L2/3 (n = 275 cells), L4 (n = 164 cells) and L5 (n = 153 cells). To test connectivity, we injected brief current pulses into each patched neuron to elicit action potentials and monitored the responses of all other simultaneously recorded neurons to identify unitary excitatory postsynaptic currents (uEPSCs, *Figure 4C*). To confirm that the recorded cells were excitatory neurons, we analyzed the firing pattern of each cell in response to sustained depolarizing current and examined the morphology of each neuron using avidin-biotin-peroxidase staining (see Methods). In addition, we measured the inter-soma distances between all pairs of simultaneously recorded neurons. Cells that did not show definitive electrophysiological and/or morphological features of excitatory neurons (5.9%, 35/592) were excluded from further analysis. In total, we tested 2049 potential excitatory connections and identified 112 synaptic connections. The uEPSCs had a latency of 2.71 ± 1.06 ms (n = 112 connections analyzed; mean ± SD), an amplitude of 12.83 ± 14.11 pA (n = 112 connections analyzed, mean ± SD), and were blocked by bath application of glutamatergic antagonists CNQX (20 μM) and APV (100 μM; uEPSC amplitude = 10.5 ± 5.4 pA and 0.0 ± 0.0 pA before and after the application of antagonists; median ±SE; n = 15 connections tested, p=6 × $10^{-5}$, Wilcoxon signed-rank test), further confirming that these were excitatory connections.

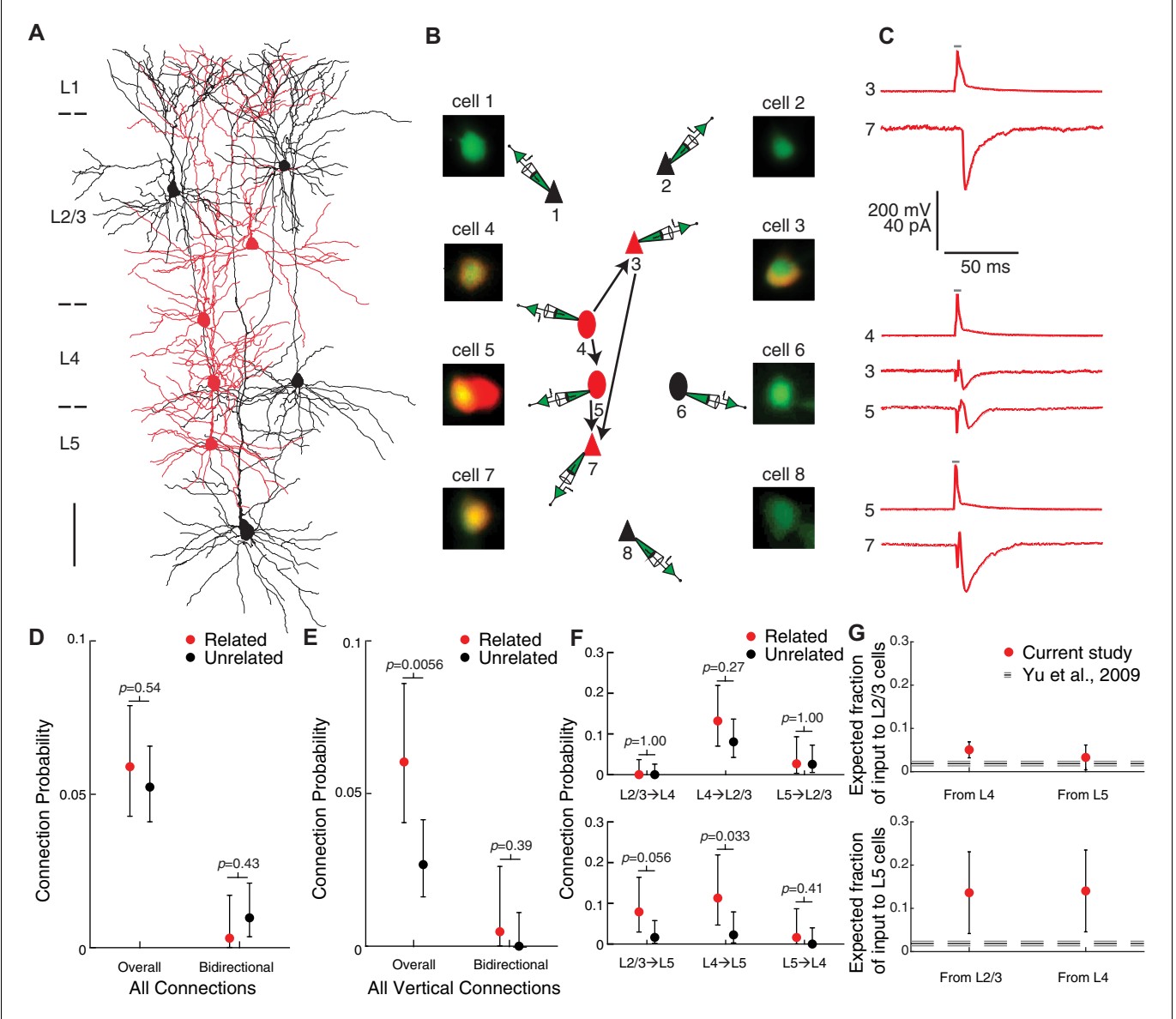

**Figure 4.** Vertical, across-layer connections are selectively increased between excitatory neurons in translaminar clones. (A–C) Example recording session from four clonally related cells (red) and four nearby, unrelated control cells (black). (A) Morphological reconstruction of all eight neurons. Scale bar, 100 μm. (B) Schematic of connections identified, as well as fluorescence images of each patched cell confirming the overlap of red (lineage tracer) and green (pipette solution) in related cells and green only in control cells. Triangles, pyramidal neurons; ovals, L4 excitatory neurons. (C) Presynaptic action potential (AP) and postsynaptic uEPSC traces for each connection (average of at least 30 trials each). Gray bar indicates period of depolarizing current injection to presynaptic neuron. (D) Connection probabilities among related and unrelated neurons, pooling all connections tested (n = 42/712 potential connections and 1/324 pairs with both directions tested for related neurons; n = 70/1337 potential connections and 6/617 pairs with both directions tested for unrelated neurons). (E) Connection probabilities among related and unrelated neurons, pooling all vertical, across-layer connections tested (n = 28/464 potential connections and 1/211 pairs with both directions tested for related neurons; n = 19/711 potential connections and 0/333 pairs with both directions tested for unrelated neurons). (F) Connection probabilities among related and unrelated neurons, for each vertical connection type tested (n = 0/98, 12/91, 2/75, 6/76, 7/62, and 1/62 potential connections for related neurons and n = 0/141, 12/149, 3/118, 2/123, 2/89, and 0/91 potential connections for unrelated neurons from L2/3 to L4, L4 to L2/3, L5 to L2/3, L2/3 to L5, L4 to L5, and L5 to L4, respectively). (G) Estimated fraction of vertical, across layer input to L2/3 cells (top panel) and L5 cells (bottom panel) coming from clonally related neurons based on our empirically measured clone sizes and connection probabilities. For comparison, the prediction based on previous work (*Yu et al., 2009*) is shown in black dashed lines. For (D–F), error bars are 95% Clopper-Pearson confidence intervals and p-values are computed using Fisher's exact test. For (G), error bars and gray dashed lines are propagated standard error of the estimates (see Methods). See also *Figure 4—Source data 1*.

The online version of this article includes the following source data for figure 4:

**Source data 1.** Summary of connectivity data, related to *Figures 4* and *5* and *Table 1*.

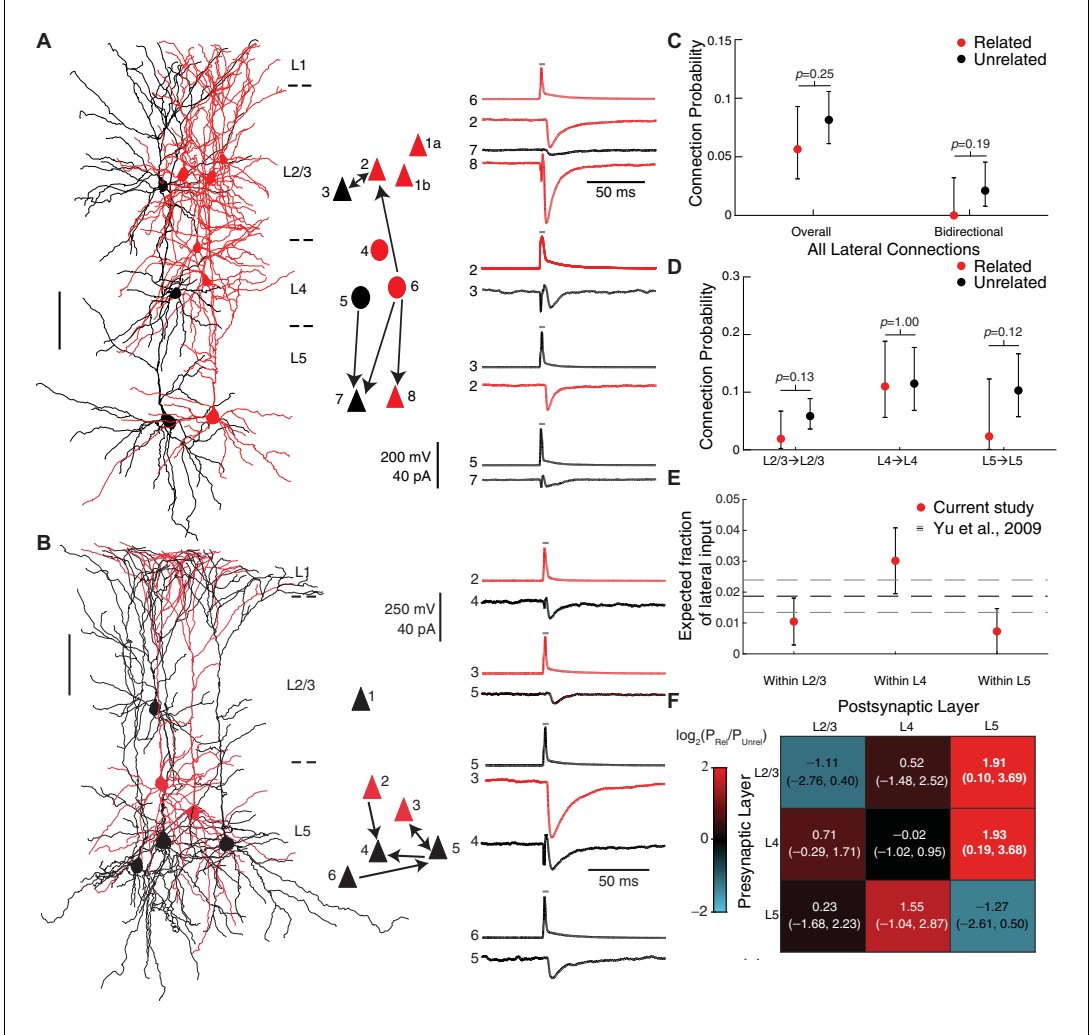

**Figure 5.** Lateral, within-layer connections are not increased between excitatory neurons in translaminar clones. (**A and B**) Example recording sessions testing within-layer connections among clonally related cells (red) and nearby, unrelated control cells (black) within L2/3 (**A**) and L5 (**B**). Scale bars, 100 μm. Triangles, pyramidal neurons; ovals, L4 excitatory neurons. Presynaptic action potential (AP) and postsynaptic uEPSC traces for each connection are an average of at least 20 trials each. Gray bar indicates period of depolarizing current injection to presynaptic neuron. (**C**) Connection probabilities among related and unrelated neurons, pooling all lateral, within-layer connections tested (n = 14/248 potential connections and 0/113 pairs with both directions tested for related neurons; n = 51/626 potential connections and 6/284 pairs with both directions tested for unrelated neurons). (**D**) Connection probabilities among related and unrelated neurons, for each lateral connection type tested (n = 2/105, 11/100, and 1/43 potential connections for related neurons and n = 20/342, 17/148, and 14/136 potential connections for unrelated neurons within L2/3, L4, and L5, respectively). (**E**) Estimated fraction of lateral inputs to a single cell within L2/3, L4, or L5 that comes from clonally related neurons based on our empirically measured clone sizes and connection probabilities. For comparison, the prediction based on previous work (*Yu et al., 2009*) is shown in black dashed lines. (**F**) Heatmap of the log ratio of the connection probabilities for related and unrelated neurons, with additive smoothing (α=1, see Methods) by connection type tested. For (**C–D**), error bars are 95% Clopper-Pearson confidence intervals and *p*-values are computed using Fisher's exact test. For (**E**), error bars and gray dashed lines are propagated standard error of the estimates (see Methods) For (**F**), the 95% confidence interval, in parentheses below the value, is computed by resampling; significant values are highlighted in bold. See also *Figure 5—figure supplements 1*, *2* and *3*.

The online version of this article includes the following figure supplement(s) for figure 5:

**Figure supplement 1.** Power analysis, related to *Figure 5*.

**Figure supplement 2.** Comparison of connectivity between clonally related neurons and distance-matched controls.

**Figure supplement 3.** Connectivity differences between clonally related and unrelated neurons in different rostrocaudal positions.

**Table 1.** Generalized linear model of connectivity.

Connectivity was modeled as a binomial response variable with the following predictors: lineage relationship (1 for related, 0 for unrelated), connection type (1 for vertical, 0 for lateral), Euclidean distance between the cells in microns, and rostrocaudal position (a numeric factor from 1 to 5; see Materials and methods). '×' denotes an interaction between two linear terms. Overall $\chi^2 = 33.5$ compared to constant model, p=2.26 × 10$^{-4}$, 1988 error degrees of freedom. The four terms with small *p*-values are: connection class (connection probability $P$ is lower for unrelated vertical connections, compared to unrelated lateral), Euclidean distance (P decreases with increasing distance for unrelated lateral connections), lineage × connection type ($P$ is higher for related vertical pairs), and connection type × Euclidean distance (the effect of Euclidean distance on $P$ depends on the type of connection tested).

| Term | Estimated coefficient | SE | t-statistic | *p*-value |
|---|---|---|---|---|
| Constant | −1.82 | 0.43 | −4.22 | 2.37·10$^{-5}$ |
| Lineage | −0.47 | 0.58 | −0.81 | 0.42 |
| Connection type | −1.55 | 0.75 | −2.06 | 0.039 |
| Euclidean distance | −8.42·10$^{-3}$ | 4.40·10$^{-3}$ | −1.91 | 0.056 |
| Rostrocaudal position | 0.074 | 0.11 | 0.65 | 0.51 |
| Lineage × Connection type | 1.25 | 0.59 | 2.11 | 0.035 |
| Lineage × Euclidean distance | 2.05·10$^{-4}$ | 2.46·10$^{-3}$ | 0.083 | 0.93 |
| Lineage × Rostrocaudal position | 0.028 | 0.15 | 0.18 | 0.86 |
| Connection type × Euclidean distance | 7.97·10$^{-3}$ | 3.99·10$^{-3}$ | 2.00 | 0.046 |
| Connection type × Rostrocaudal position | 5.86·10$^{-3}$ | 0.20 | 0.030 | 0.98 |
| Euclidean distance × Rostrocaudal position | −5.97·10$^{-4}$ | 9.33·10$^{-4}$ | −0.64 | 0.52 |

To determine the relationship between cell lineage and connection probability (P), we compared pairs consisting of two labeled cells within the same clone (i.e. 'related' pairs) to pairs consisting of one labeled and one unlabeled cell (i.e. 'unrelated' pairs). Pairs consisting of two unlabeled cells were not included as controls, since we could not be certain that those pairs are unrelated (i.e., they could be related, but their progenitor was not labeled). Overall, there was no evidence for a difference in connectivity between related and unrelated pairs (P=5.9% and P=5.2% for related and unrelated pairs, respectively; p=0.54, Fisher's exact test; *Figure 4D*). A single bidirectional connection was identified between related neurons (1/324, P=0.31%), and six were identified between unrelated neurons (6/617, P=0.97%; p=0.43, Fisher's exact test, *Figure 4D*).

However, when we considered only connections linking cells vertically, across layers, we found that connection probability was increased between related pairs compared to unrelated pairs (P=6.0% and P=2.7% for related and unrelated pairs, respectively; p=0.0056, Fisher's exact test; *Figure 4E*), consistent with prior studies (*Yu et al., 2009*; *Yu et al., 2012*). A single vertical bidirectional connection was identified between related neurons (P=0.47%), and none were identified between unrelated neurons (P=0.0%; p=0.39, Fisher's exact test; *Figure 4E*). At the level of layer-specific connection types, we found that the connection probabilities from L4 to L5 (P=11.3% and P=2.3% for related and unrelated pairs, respectively; p=0.033, Fisher's exact test; *Figure 4F*) and from L2/3 to L5 (P=7.9% and P=1.6%, for related and unrelated pairs respectively; p=0.056, Fisher's exact test; *Figure 4F*) were higher between related compared to unrelated pairs. If we assume that only a shared radial glial lineage predisposes cells to be synaptically connected, and given that we had on average N = 2 radial glial progenitors per clone (*Figure 1G–I*), then the vertical connection probability between neurons derived from a single radial glial cell would be estimated to be even higher (approximately P=9.4%) in order to yield the P=6.0% vertical connectivity that we found among clonally related cells (see Methods).

To estimate the contribution of clonally related neurons as a fraction of the total input a cell receives, we used a simple quantitative model of connectivity based on our empirically measured connection probabilities and clone sizes. Briefly, we modeled the number of input connections to a particular cell from both related and unrelated cells in different layers of a cortical column as a binomial distribution with the connection probabilities set to the empirically measured connection probabilities (see Methods for additional assumptions). If we fit the model using previously reported

connection probabilities (P=36.3%, 65/179, for related pairs; P=6.3%, 9/143, for unrelated pairs; *Yu et al., 2009* Figure 3H, pooling connections tested across control groups) and assume a clone size of 6 neurons using the retroviral lineage tracing method with labeling at E12.5 (*Yu et al., 2009*), the model predicts that, despite the very high connection probability between closely related 'sister' cells, only a small fraction of the total input to a cell comes from its sister cells (1.9 ± 0.5%, mean ± SEM, *Figure 4G*). Given the much larger clone sizes in our study, we wondered whether our data would suggest a larger fraction of input coming from related neurons, and whether specific layer-defined connection types would show a substantial increase in the fraction of input from clonally related cells. Indeed, using our empirically measured connection probabilities and clone size (60 neurons on average), the model estimates that a larger fraction of inputs from L4 to L2/3, L4 to L5, and L2/3 to L5 originates from cells with a common developmental lineage (5.0 ± 1.8% of L4→L2/3 connections, 14.0 ± 9.5% of L4→L5 connections, and 13.6 ± 9.4% of L2/3→L5 connections; estimate ± SE, *Figure 4G*). These findings show that certain layer-specific, vertical connections show a stronger contribution of lineage-associated connections than expected based on previous data, and suggest that even distantly related cells derived from multiple radial glia develop preferential connectivity, possibly as a result of physical proximity during neurogenesis and/or a common migratory route.

## Lateral, within-layer connections are not increased between excitatory neurons in translaminar clones

In contrast to vertical connections, we did not find evidence for an increase in the number of lateral connections within the same cortical layer between related neurons compared to unrelated neurons (P=5.6% and P=8.1% for related and unrelated pairs, respectively; p=0.25, Fisher's exact test; *Figure 5A–C*). There was also no statistically significant difference in bidirectional lateral connections between related and unrelated pairs (P=0% for related lateral pairs and P=2.1% for unrelated lateral pairs in which both directions of connectivity were tested; p=0.19, Fisher's exact test; *Figure 5C*). If anything, related neurons were less likely to be connected to each other within L2/3 and within L5 (P=1.9% and P=2.3% for related pairs within L2/3 and L5, respectively) compared to unrelated pairs (P=5.9% and P=10.3% for unrelated pairs within L2/3 and L5, respectively) although the differences were not statistically significant (p=0.13 and p=0.12 for L2/3 and L5, respectively, Fisher's exact test; *Figure 5D*). Our connectivity model estimates that only a small fraction of lateral, within-layer input to a cell comes from related neurons (1.0 ± 0.8% of connections within L2/3, 3.0 ± 1.1% of connections within L4, and 0.7 ± 0.7% of connections within L5; estimate ± SE; *Figure 5E*).

Importantly, given that our translaminar clones are derived from two radial glia on average (range 1–4, *Figure 1G–J* and *Figure 1—figure supplement 2*), we performed power analysis over a wide range of possible effect sizes and confounding factors that might result from pooling multiple radial glial lineages (see Methods and *Figure 5—figure supplement 1*). We found that, if we assume that the increase in connectivity among neurons derived from a single radial glial cell is the same for vertical and lateral connections, and that there are two radial glial lineages per clone, we had approximately 97% statistical power to detect an increase in lateral connectivity of the same magnitude as we found for vertical connectivity (*Figure 5—figure supplement 1B*). Indeed, because of the large number of connections tested, our statistical power is 80% or greater as long as the number of radial glial lineages per clone is less than 6 (*Figure 5—figure supplement 1B*). Therefore, these data convincingly show that clonally related neurons are not preferentially connected to each other within a cortical layer, even if one considers only neurons derived from the same radial glial cell as meaningfully 'related'. These findings suggest a revised model of lineage-associated synaptic connectivity that is connection type-specific, with related cells preferentially connected vertically, across cortical layers, but not laterally within the same layer (*Figure 5F*).

Since connection probability also depends on the distance between cells (*Perin et al., 2011*; *Ko et al., 2011*), and potentially on cortical area, we performed additional analyses in order to take these variables into account. First, for each pair of clonally related neurons we identified a set of distance-matched control pairs with the same pre- and post-synaptic layers and with the same (less than 20 μm difference) tangential and vertical distances between the cells. We compared connectivity rates between related pairs and distance-matched control pairs by bootstrapping (see Methods). We found similar results as described above, with increased vertical connection probability between related neurons and no evidence for a difference in lateral connection probability (*Figure 5—figure*

*supplement 2*). Second, we sorted the data into two groups according to the rostro-caudal position of each clone, which revealed similar changes in connectivity between related and unrelated neurons in both groups (*Figure 5—figure supplement 3*). Third, we built a generalized linear model of connection probability taking into account cell lineage (related or unrelated), connection type (vertical or lateral), Euclidean distance between cells and rostro-caudal position as predictors (see Methods and *Table 1*). The model revealed a significant interaction between cell lineage and connection type (p=0.035), further supporting that the effect of cell lineage on connectivity depends on the type of connection tested. The model also revealed a weak decrease in connection probability with Euclidean distance for lateral connections (p=0.056) and no evidence for any influence of rostro-caudal position (p>0.5 for the main effect and all interactions).

## Discussion

In summary, we show that translaminar clones labeled at E10.5 in the mouse neocortex are composed of a diverse ensemble of excitatory neurons. In addition, we show that these distantly related excitatory neurons, derived from two radial glial lineages on average, are preferentially connected vertically, across cortical layers, but not laterally, within the same cortical layer. These findings carry both mechanistic and functional implications regarding circuit assembly in the neocortex, and suggest that integration of vertical input from related neurons with lateral input from unrelated neurons may represent a fundamental principle of cortical information processing that is initially established by hardwired developmental programs.

### Cell type composition of translaminar clones

Nearly all of the clones labeled at E10.5 in our study spanned cortical layers 2–6, consistent with prior studies labeling progenitors at this early developmental stage (*Tan et al., 1998*; *Kaplan et al., 2017*). Some studies have reported upper layer fate-restriction among a subset of progenitors present at E10.5 (*Franco et al., 2012*; *Franco and Müller, 2013*; *Gil-Sanz et al., 2015*) using the *Cux2*-CreER driver line but not the *Nestin*-CreER line (*Franco et al., 2012*). It is possible that *Cux2*-CreER labels a small subset of radial glia that exhibit upper layer-restricted neurogenesis potential, or that the progenitor pool labeled using our *Nestin*-CreER driver is distinct from the *Cux2*-CreER-positive fraction. In contrast, we observed that approximately one-fifth of *Nestin*-CreER clones labeled at E11.5 were restricted to cortical layers 2–4, which is consistent with the presence of a subset of upper layer-restricted radial glial cells that emerge sometime between E10.5–E11.5. We did not observe any definite deep-layer restricted clones or other obvious layer-restricted patterns that have been described when labeling progenitors at later developmental stages (*Llorca et al., 2019*). Given the low doses of tamoxifen administered in our study, we may be biased to label a potentially nonrandom subset of progenitors with the highest *Nes* promoter activity at E10.5, and so it is possible that our clones are not representative of the entire progenitor population. However, our observation that these clones generate nearly the full diversity of cell types present within the local cortical areas examined (*Figure 3B* and S5B) argues against this. Interestingly, the only transcriptomic subtype that appeared relatively underrepresented in our clones was a subset of L2/3 intratelencephalic neurons (L2/3 IT VISp Rrad; *Figure 3B*). Further work is needed to determine whether this particular subtype of L2/3 neurons may arise primarily from *Cux2*-positive or other progenitors that are not enriched in our progenitor pool.

A recent study found that approximately 10% of clones derived from *Nestin*-positive progenitors labeled at E12.5 are restricted to either the superficial or deep layers, and nearly a quarter of the labeled translaminar clones were composed exclusively of corticocortical projection neurons (*Llorca et al., 2019*). We did not observe any bias in either the broad class (i.e. intratelencephalic versus corticothalamic versus pyramidal tract) or specific transcriptomic cell types of neurons within our translaminar clones (*Figure 3C,D* and *Figure 3—figure supplement 1C,D*). However, that study used a different Cre driver line (*Emx1*-CreERT2) which may target a different progenitor pool than the one described in our study and labeled progenitor later in development. Given that many of our clones were derived from two or more radial glial lineages (range 1–4, *Figure 1G–J* and *Figure 1—figure supplement 2*) we cannot exclude the possibility that individual radial glial lineages are fate-restricted to give rise to particular layers. Thus, our findings do not contradict those reported in *Llorca et al. (2019)*.

Recent single-cell RNA-sequencing studies have shown that excitatory neuron cell types in adult mice are largely region-specific (*Tasic et al., 2018*; *Saunders et al., 2018*). Here we profiled the transcriptomes of cells from two primary sensory areas, V1 and SS1, and found that the vast majority of cells from both regions map to V1-specific transcriptomic cell types rather than ALM-specific transcriptomic cell types, suggesting that cell types in different primary sensory areas are more similar to each other than they are to cell types in motor cortex. Interestingly, there were still subtle region-specific differences in gene expression between V1 and SS1 that could be detected using a generalized linear model, suggesting that these two primary sensory areas may also be composed of distinct excitatory neuron subtypes. Importantly, there was no evidence to suggest that the quality of our mapping to the V1/ALM reference dataset was worse for SS1 cell than for V1 cells; however if a reference atlas for SS1 becomes available in the future it would be interesting to re-examine our data to better characterize the developmental timeline of area-specific gene expression signatures in these two primary sensory areas.

## Connectivity matrix of translaminar clones

It has been previously reported that clonally related excitatory neurons are more likely to be synaptically connected (*Yu et al., 2009*; *Yu et al., 2012*; *He et al., 2015*), and the prevailing view has been that clonally related neurons form highly and uniformly interconnected subnetworks within the neocortical circuit (*Li et al., 2018*). In contrast to these prior studies, we find that connectivity between excitatory neurons in translaminar clones is connection type-specific. In particular, clonally related neurons are more likely to be synaptically connected vertically, across cortical layers, but not laterally, within the same cortical layer. Our findings challenge the current model and suggest an updated view of lineage-associated circuit assembly in which developmental programs promote integration of vertical inputs from related neurons with lateral inputs from unrelated neurons.

A recent study using chimeric mice in which fluorescently labeled induced pluripotent stem cells (iPSCs) were injected into blastocysts at E3.5 has examined lateral connections between related cells within L4 and found a transient increase in synaptic connectivity at P13-P16, followed by an increase in the fraction of connections that are reciprocal, but not one-way, at P18-P20 (*Tarusawa et al., 2016*). We did not find any evidence for an increase in either overall connectivity or bidirectional connectivity within L4; however, it is possible that if the increase in connectivity in L4 is transient and only present from P13-P16 for overall connectivity and from P18-P20 for bidirectional connectivity that we may not have detected it, as our data predominantly span the space across these time windows from P15-P20. Another possibility is that the iPSC-derived neurons may have altered synaptogenesis and connectivity rates due to chromosomal instability and altered gene expression programs of iPSCs (*Mayshar et al., 2010*). Additional experiments to explore the possibility of a transient increase in lateral connections among related neurons within L4 that include a direct comparison between iPSC-derived clones and clones labeled using other methods may help to resolve this issue.

Our finding that clonally related neurons are only rarely connected by lateral connections within L2/3 is quite unexpected given prior studies showing similar feature selectivity between clonally related neurons in this layer (*Li et al., 2012*), even in very large clones (*Ohtsuki et al., 2012*). Several studies have shown that excitatory neurons in L2/3 with similar orientation tuning are more likely to be synaptically connected and have stronger synapses compared to cells with dissimilar tuning preferences (*Ko et al., 2011*; *Cossell et al., 2015*). Thus, the proposed model (*Li et al., 2018*) has been that increased connections between clonally related neurons within L2/3 underlies their similarity in tuning. However, we found no increase in lateral connections between related cells in L2/3, suggesting that an alternate mechanism such as common input from L4, from nearby unrelated neurons, or from long-range feedback connections may drive their similarity in orientation tuning preferences. These results suggest a novel functional role of lateral connections within L2/3 in facilitating communication between adjacent developmental clones.

Our results highlight L5 as a potential hub within translaminar clones, with the most striking increases in connectivity seen in the projections from layers L2/3 and L4 to L5. Neurons in L5 serve as a major output of the cortex with important roles in integrating feedback from higher cortical areas and in top-down modulation by brain states (*Kim et al., 2015*), and altered gene expression in deep layer neurons during midfetal development has been implicated in neuropsychiatric disorders such as autism (*Willsey et al., 2013*). Our data suggest that integration of translaminar input from

related neurons with intralaminar input from unrelated neurons in L5 may represent an organizing principle of lineage-associated circuit assembly. While L5 has traditionally been less amenable to in vivo functional studies given its depth, recent advances in calcium imaging such as three-photon microscopy and genetically encoded calcium indicators (*Ouzounov et al., 2017*) may enable functional analysis of cortical computation in vivo simultaneously in superficial and deep layers of translaminar clones. Future studies aimed at dissecting the functional role of lineage-associated synaptic connectivity across the cortical column may provide mechanistic insight into abnormal circuit function in neuropsychiatric disease.

The mechanism by which translaminar clones of excitatory neurons form specific connections is thought to involve gap junction coupling during migration along the same radial glial fiber (*Yu et al., 2012*; *He et al., 2015*). A recent study has further shown that it is the coupling between clonally related neurons, and not between the postmitotic neurons and their radial glia, that promotes specific synapse formation between radially aligned sister neurons (*He et al., 2015*). Furthermore, this coupling requires the inside-out migration of related neurons along a similar path and is abolished by removal of REELIN or its downstream effector DAB1 (*He et al., 2015*). Our finding that connections from L2/3 to L5 and from L4 to L5 are most strongly enhanced between clonally related neurons is consistent with a mechanism that requires inside-out migration along a radial glial fiber and further suggests that as migrating neurons travel to reach the more superficial layers, their axons may somehow 'stick' to the maturing apical dendrites of the deep layer neurons they are passing. Furthermore, our findings suggest that even distantly related neurons that are in close physical proximity and migrate along the same radial glial pathway may be predisposed to develop vertical connections with one another, and not just closely related neurons derived from the same radial glial cell. Interestingly, early studies in the rat suggested that migration of clonally related neurons along multiple radial glial fibers may be quite common (*Walsh and Cepko, 1988*), resulting in substantial tangential dispersion that requires EphA/Ephrin-A signaling during neuronal migration (*Torii et al., 2009*). Expression of specific cell adhesion molecules may also play a role in promoting vertical connections and/or repelling lateral connections among clonally related neurons (*Tarusawa et al., 2016*). For example, it is possible that clonally related neurons residing in the same layer are derived from the same intermediate progenitor cell, and that this close lineage relationship confers an additional reduction in connectivity (*Ellender et al., 2019*) which is not inherited by neurons in other layers derived from the same radial glial cell. Future studies utilizing temporally evolving lineage tracing methods (*Chan et al., 2019*), and studies dissociating cell lineage from migratory path, could provide further insights into the molecular mechanisms underlying the selective formation of vertical connections among related neurons.

Our study is well-powered to detect a universal increase in connectivity between clonally related neurons, and the virtual absence of lateral connections observed between related neurons in our study cannot be explained by possible confounding of the measured connection probability due to pooling of multiple radial glial lineages (*Figure 5—figure supplement 1*). Nonetheless, it is possible that with additional sampling, other differences in connectivity may emerge in specific layer-defined connection types or transiently at certain developmental time points. Layer six neurons were not included in our study for two main reasons: 1) L6 neurons undergo nuclear translocation, whereby they attach their leading process directly to the pial surface, rather than locomoting along a radial glial fiber (*Gupta et al., 2002*) and therefore may be less likely to show radial glial fiber-dependent connectivity; and 2) including an additional layer in our study would have disproportionately increased the number of possible connection types we needed to test (from 9 to 16), making the task experimentally impractical. Of note, while we are well-powered to detect an increase in connectivity, our statistical power to detect a *decrease* in connectivity is substantially lower due to the overall low connectivity rate among excitatory neurons (we estimate a 16% probability of detecting a 50% decrease in connectivity, *Figure 5—figure supplement 1B*). Thus, our data cannot exclude the possibility that lateral connectivity is actually decreased between clonally related neurons.

Similarly, since we focused on V1 and SS1, both primary sensory areas, it is possible that a different pattern of connectivity among clonally related neurons is present in other cortical areas such as primary motor cortex. Our acute slice preparation only allows us to test local connections; however, given that transcriptomic cell type correlates with the long-range projection pattern of excitatory neurons (*Tasic et al., 2018*), our finding that individual clones contain multiple diverse transcriptomic types suggests that they also project to diverse targets. Additional experiments using different

methods for lineage tracing and connectivity profiling, focusing on different brain regions and using adult animals, will be necessary to determine the generalizability of the connectivity pattern we describe here and delineate the long-range inputs and outputs of clonal units. However, our data suggest that the integration of feedforward, intra-clonal input with lateral, inter-clonal information may represent a developmentally programmed connectivity motif for the assembly of neocortical circuits.

# Materials and methods

## Key resources table

| Reagent type (species) or resource | Designation | Source or reference | Identifiers | Additional information |
|---|---|---|---|---|
| Gene *Mus musculus* | nestin | | MGI:101784 NCBI Gene: 18008 | |
| Strain, strain background *Mus musculus* | C57Bl/6J | The Jackson Laboratory | JAX Stock no. 000664 RRID:IMSR_JAX:000664 | Females |
| Strain, strain background *Mus musculus* | CD1 | Obtained from the Baylor College of Medicine Center for Comparative Medicine | | |
| Strain, strain background *Mus musculus* | Nestin-CreER | Obtained from Dr. Mirjana Maletic-Savatic Lab at Baylor College of Medicine | | Cryopreserved by Andreas Tolias Lab at Baylor College of Medicine |
| Strain, strain background *Mus musculus* | Ai9 | The Jackson Laboratory | JAX Stock no. 007909 RRID:IMSR_JAX:007909 | |
| Antibody | Rabbit anti-Tbr2 | Abcam | cat. no. AB23345 RRID:AB_778267 | |
| Antibody | Mouse anti-Pax6 | Developmental Studies Hybridoma Bank (DSHB) at the University of Iowa. PAX6 was deposited to the DSHB by Kawakami, A. | DSHB Hybridoma Product PAX6 RRID:AB_528427 | |
| Antibody | Goat anti-tdTomato | Sicgen | cat. no. AB8181-200 RRID:AB_2722750 | |
| Chemical compound, drug | Tamoxifen (≥99%) | Sigma-Aldrich | cat. no. T5648 | |
| Chemical compound, drug | Progesterone (≥99%) | Sigma-Aldrich | cat. no. P0130 | |
| Other | Corn Oil | Sigma-Aldrich | cat. no. C8267 | |
| Other | DNA-OFF | Clontech | Cat. no 9036 | |
| Other | RNase Zap | Thermo Fisher Scientific | Cat. no. AM9780 | |
| Peptide, recombinant protein | Recombinant RNase inhibitor | Clontech | 2313A | |
| Other | Potassium D-gluconate (K-gluconate,≥99%) | Sigma-Aldrich | cat. no 64500 | |
| Other | Potassium chloride (KCl, for molecular biology,≥99.0%) | Sigma-Aldrich | cat. no. P9541 | |
| Other | HEPES solution (1 M, BioReagent) | Sigma-Aldrich | cat. no. H3537 | |

*Continued on next page*

*Continued*

| Reagent type (species) or resource | Designation | Source or reference | Identifiers | Additional information |
|---|---|---|---|---|
| Other | Ethylene glycol-bis (2-aminoethylether)-$N,N,N',N'$-tetraacetic acid (EGTA, for molecular biology, $\geq$97.0%) | Sigma-Aldrich | cat. no. E3889 | |
| Other | Adenosine 5'-triphosphate magnesium salt (Mg-ATP, $\geq$95%) | Sigma-Aldrich | cat. no. A9187 | |
| Other | Guanosine 5'-triphosphate sodium salt hydrate (Na-GTP, $\geq$90%) | Sigma-Aldrich | cat. no. 51120 | |
| Other | Phosphocreatine disodium salt hydrate (Na$_2$-phosphocreatine, $\geq$97%) | Sigma-Aldrich | cat. no. P7936 | |
| Other | Glycogen (RNA grade) | Thermo Fisher Scientific | cat. no. R0551 | |
| Other | Biocytin ($\geq$98%) | Sigma-Aldrich | cat. no. B4261 | |
| Sequence-based reagent | ERCC RNA spike-in mix | Thermo Fisher Scientific | cat. no. 4456740 | sequences available at website |
| Other | Tris-EDTA buffer solution (TE buffer, BioUltra, for molecular biology) | Sigma-Aldrich | cat. no. 93283 | |
| Other | Triton X-100 | Sigma-Aldrich | cat. no. T8787 | |
| Other | Betaine (BioUltra, $\geq$99.0%) | Sigma-Aldrich | cat. no. 61962 | |
| Other | dNTPs (25 mM each) | Thermo Fisher Scientific | cat. no. R1121 | |
| Peptide, recombinant protein | Superscript II reverse transcriptase (SSIIRT) | Thermo Fisher Scientific | cat. no. 18064014 | |
| Other | MgCl$_2$ (1M, molecular biology grade) | Thermo Fisher Scientific | cat. no. AM9530G | |
| Commercial assay, kit | KAPA Biosystems HiFi HotStart Ready Mix | Thermo Fisher Scientific | cat. no. NO0295239 | |
| Other | TAPS ($\geq$99.5%) | Sigma-Aldrich | cat. no. T5130 | |
| Other | Polyethylene glycol solution (PEG-8000, 40% wt/vol) | Sigma-Aldrich | cat. no. P1458 | |
| Commercial assay, kit | Nextera XT index kit v2 set A for 96 indices, 384 samples | Illumina | cat. no. FC-131–2001 | |
| Commercial assay, kit | Axygen AxyPrep mag PCR clean-up kit | Thermo Fisher Scientific | cat. no. 14223151 | |
| Commercial assay, kit | KAPA HiFi PCR Kit | KAPA Biosystems | cat. no. KK2103 | |
| Other | Hoechst 33342 | Invitrogen | cat. no. H3570 | |
| Other | Antigen unmasking solution | Vector Laboratories | cat. no. H-3300 | |
| Other | Vectashield antifade mounting medium | Vector Laboratories | cat. no. H-1000 | |

*Continued on next page*

*Continued*

| Reagent type (species) or resource | Designation | Source or reference | Identifiers | Additional information |
|---|---|---|---|---|
| Sequence-based reagent | Oligo-dT30VN (HPLC purified) | Biomers.net | oligonucleotide | 5'- AAGCAGTGGTATCAA CGCAGAGTAC TTTTTTTTTTTTTTTTTT TTTTTTTTTTTTVN-3', where V represents A, C or G and N represents any nucleotide |
| Sequence-based reagent | IS PCR Oligo (HPLC purified) | Biomers.net | oligonucleotide | 5'-AAGCAGTGGTATCA ACGCAGAGT-3' |
| Sequence-based reagent | Template switching oligonucleotide (LNA-TSO; RNase-free; HPLC purified | Exiqon | oligonucleotide | 5'-AAGCAGTGGTATCAACGCA GAGTACrGrG+G-3', where rG indicates riboguanosines and +G indicates a locked nucleic acid (LNA)-modified guanosine |
| Software, algorithm | Patchmaster software | HEKA | RRID:SCR_000034 | |
| Software, algorithm | STAR v2.4.2a | https://github.com/alexdobin/STAR/releases/tag/STAR_2.4.2a | RRID:SCR_015899 | |
| Software, algorithm | Neurolucida | MBF Bioscience | RRID:SCR_001775 | |
| Other | Glass capillaries (2.0 mm OD, 1.16 mm ID) | Sutter Instruments | | |

## Animals

All experiments were carried out in accordance with, and with approval from, the Institutional Animal Care and Use Committee (IACUC) at Baylor College of Medicine (protocol #AN-4703). The Nestin-CreER line was obtained from M. Maletic-Savatic (BCM) and maintained in A. Tolias' laboratory by crossing heterozygous males with wild type C57Bl/6J females. Each generation, potential stud males were crossed with a reporter line to confirm the lack of transgene expression in the absence of tamoxifen administration, and only those males showing minimal to no 'leaky' recombination in the P1 offspring of this test cross were used as breeders for maintaining the CreER line. The Nestin-CreER line is cryopreserved at BCM for potential future use. The reporter line ROSA26-CAG-LSL-tdTomato-WPRE (Ai9) was acquired from the Jackson Laboratory (JAX Stock #007909). The outbred CD1 line was obtained from the Center for Comparative Medicine at BCM. Ten mice were used for quantification of clone size and width (3 males and 3 females, 4 unknown), 9 mice (all males) were used for Patch-seq experiments, and 43 mice (26 males, 7 females, 10 uncertain) were used for electrophysiology experiments. For clone quantification, animals were sacrificed at postnatal day 10 (P10) and for Patch-seq and multi-patching experiments animals were sacrificed at P15-P20. All animals were on the C57Bl/6J or mixed C57Bl/6J; CD1 genetic background and were group housed with their littermates and foster mothers (both CD1 and C57Bl/6J foster mothers were used) on a 12 hr light-dark cycle.

## Lineage tracing method

We used a tamoxifen-inducible Cre-lox transgenic approach for lineage tracing similar to previous studies (*Gao et al., 2014*). Two breeding strategies were used: The majority of experimental animals were generated by crossing *Nestin*-CreER heterozygous males with Ai9 homozygous females (both on a C57Bl/6J background). A subset of experimental animals was generated by crossing double homozygous Cre; Ai9 males (C57/Bl6J background) with wild type CD1 females. The latter breeding strategy increased the yield of experimental animals per litter and negated the need for genotyping of the pups (all are double heterozygotes).

Extensive precautions were taken to minimize the variability in timing of pregnancies across litters. Males were housed with females overnight for a maximum of 15 hr (5 pm to 8 am). The day after mating was designated as E0.5. Females were weighed daily after each cross, and a gain of more than ~3 gm overall since E0.5 and more than ~1 gm in the preceding 24 hr were used as criteria to assess pregnancy status and determine which females would receive tamoxifen treatment at E9.5, E10.5 or E11.5. Application of these criteria rarely missed any pregnancies, but occasionally resulted in false positives (females that were treated but turned out not to be pregnant). In particular, weight gain due to pregnancy at E9.5 could be difficult to distinguish from nonspecific fluctuations in baseline weight especially in in older/heavier females. Other approaches such as plugging were also used, but in our hands were not as reliable as the trend in daily weights. Females that were not found to be pregnant could be used in additional rounds of crosses.

Tamoxifen and progesterone were dissolved together in corn oil and administered to pregnant dams at E9.5, E10.5 or E11.5 at a dose of 40–50 and 20–25 mg/kg, respectively, by orogastric gavage. To minimize variability in labeling across litters, tamoxifen was typically administered in the mid to late morning of the designated treatment day (E9.5, E10.5, or E11.5). To help prevent tamoxifen-induced pregnancy loss (*Milligan and Finn, 1997*), pregnant mice also received 2 mg of progesterone dissolved in corn oil subcutaneously twice a day (morning and evening), beginning the day after tamoxifen treatment and continuing until the pups were delivered by Caesarian section on E19.5 (as described in *Nagy et al., 2006*). The pups were raised by a foster mother and standard genotyping protocols were used to identify double heterozygous animals carrying both the Cre and reporter alleles if needed depending on the breeding strategy.

## Histology and immunohistochemistry

For examination of clones at postnatal day 10 (P10, *Figure 1B–F* and *Figure 1—figure supplement 1*), animals were deeply anesthetized with isoflurane and transcardially perfused with 0.1M phosphate buffered saline (PBS) followed by 4% paraformaldehyde in PBS. Fixed brains were coronally sectioned at 100 µm on a vibratome (Leica VT1000S) and stained with DAPI (0.25 µg/mL) for 10–15 min before mounting on charged glass slides with anti-fade mounting solution (1 mg/ml ρ-phenylenediamine in 90% glycerol, 10% PBS, pH ~8.0). Confocal image stacks were taken on a Zeiss LSM 510 Meta, Zeiss LSM 780, or Leica SP8 confocal microscope.

For examination of clones at E12.5 (*Figure 1G–J* and *Figure 1—figure supplement 2*), embryos were immersion fixed in 4% paraformaldehyde (PFA) for 48 hr, cryoprotected in 15% and 30% sucrose and embedded in OCT (Tissue-Tek, Sakura FineTechnical, Japan). Brains were sectioned at 20 µm thickness using a cryostat. Immunohistochemistry was performed using standard procedures. Briefly, sections were subjected to heat-mediated antigen retrieval (Antigen unmasking solution, Vector Laboratories, Burlingame, CA) at 80°C for 30 min followed by blocking in 3% Bovine Serum Albumin (BSA) containing 0.3% Triton X-100 for 1 hr. The sections were incubated overnight with primary antibodies diluted in blocking solution at 4°C (rabbit anti-Tbr2, 1:1000,; mouse anti-Pax6, 2.5 µg/mL; goat anti-tdTomato, 1:500). The following day, sections were incubated with secondary antibodies labeled with Alexa fluor 488, 555 and 647 (Molecular Probes, Eugene, OR). Sections were counterstained with Hoechst stain and mounted in Vectashield mounting medium. Images were taken using a Zeiss LSM 880 confocal microscope and assembled in Adobe Illustrator.

## Acute brain slice preparation

Acute brain slices were prepared as previously described (*Jiang et al., 2015*). In brief, animals (P15–P20) were deeply anesthetized with 3% isoflurane and decapitated. The brain was quickly removed and placed into cold (0–4°C) oxygenated physiological solution containing (in mM): 125 NaCl, 2.5 KCl, 1.25 $NaH_2PO_4$, 25 $NaHCO_3$, 1 $MgCl_2$, 25 dextrose, and 2 $CaCl_2$, pH 7.4. Parasagittal slices 300 µm thick were cut from the tissue blocks using a microslicer (Leica VT 1200). The slices were kept at 34.0 ± 0.5°C in oxygenated physiological solution for ~0.5–1 hr before recording. During the recordings, the slices were submerged in a chamber and stabilized with a fine nylon net attached to a platinum ring. The recording chamber was perfused with oxygenated physiological solution. The half-time for the bath solution exchange was 1–2 min. All antagonists were bath applied.

## Patch-seq sample collection

To obtain transcriptome data from individual neurons within translaminar clones, we used our previously described Patch-seq method (Cadwell et al., 2016; Cadwell et al., 2017). Briefly, the following modifications were made to the standard whole-cell patch-clamp workflow to improve RNA yield from patched cells. Glass capillaries were autoclaved prior to pulling patch pipettes, all work surfaces and micromanipulator pieces were thoroughly cleaned with DNA-OFF and RNase Zap, and all solutions that would come into contact with RNA were prepared using strict RNAse-free precautions. Recording pipettes of 2–4 MΩ were filled with a small volume (approximately 0.3 µl) of internal solution containing: 123 mM potassium gluconate, 12 mM KCl, 10 mM HEPES, 0.2 mM EGTA, 4 mM MgATP, 0.3 mM NaGTP, 10 mM sodium phosphocreatine, 20 µg/ml glycogen, 13 mM biocytin, and 1 U/µl recombinant RNase inhibitor, at pH ~7.25. RNA was collected at the end of whole-cell recordings by applying light suction while observing the cell under differential interference contrast (DIC) until the cell was visibly shrunken or could no longer tolerate suction. If any extracellular contents were observed to enter the pipette under DIC, the sample was discarded. Otherwise, the contents of the pipette were ejected into and RNase-free PCR tube containing 4 µl of lysis buffer consisting of 0.1% Triton X-100, 5 mM (each) dNTPs, 2.5 µM Oligo-dT$_{30}$VN, 1 U/µl RNase inhibitor, and $1 \times 10^{-5}$ dilution of ERCC RNA Spike-In Mix. cDNA synthesis, library preparation and sequencing.

Single cell RNA was converted to cDNA following the Smart-seq2 protocol (Picelli et al., 2014a; Cadwell et al., 2017). Samples were denatured at 72°C for 3 min and then 5.70 µl of RT mix was added to each sample, with final concentrations as follows: 1 × Superscript II first strand buffer, 1M Betaine, 10 U/µl SSIIRT, 5 mM DTT, 1 U/µl RNase inhibitor, 1 µM LNA-TSO, and 6 mM MgCl$_2$. The RT reaction was run at 42°C for 90 min followed by ten cycles of 50°C for 2 min, 42°C for 2 min, and the enzyme was inactivated by holding at 70°C for 15 min.

The full-length cDNA was amplified by adding 15 µl of PCR mix to each sample, with a final concentration of 1 × KAPA HiFi HotStart Ready Mix and 0.1 µM IS PCR primers, and running the following PCR program: 98°C for 3 min; 18 cycles of 98°C for 20 s 67°C for 15 s 72°C for 6 min; and 72°C for 5 min. The PCR product was purified using Axygen AxyPrep mag PCR beads according to the manufacturer's instructions but using a bead:sample ratio of 0.7:1 (17.5 µl of beads: 25 µl sample).

To construct the final sequencing libraries, we diluted each sample to a concentration of ~50 pg/µl and added 4 µl of tagmentation mix to 300 pg (6 µl) of full-length cDNA for a final concentration of: 1 × tagmentation buffer (1 mM TAPS-NaOH, 5 mM MgCl$_2$), 10% (wt/vol) PEG-8000, and 1.25 µM in-house produced Tn5 transposase (Picelli et al., 2014b; Cadwell et al., 2017). The tagmentation reaction was run in a thermal cycler at 55°C for 8 min and the Tn5 transposase was stripped by adding 2.5 µl of 0.2% (wt/vol) SDS to each sample by incubating at room temperature for 5 min. Amplification of the adapter ligated fragments was performed by adding 2.5 µl each of index 1 (N7XX) and index 2 (N5XX) primers, diluted 1:4, from the Nextera XT index kit with a unique combination of indices for each sample, as well as 5 µl of 5 × KAPA HiFi Buffer, 0.75 µl of KAPA dNTP mix (10 nM each), 1.25 µl of nuclease-free water, and 0.5 µl of KAPA enzyme (1 U/µl) for a total volume of 25 µl. The enrichment PCR was run according to the following program: 72°C for 3 min, 95 for 30 s, 12 cycles of 95°C for 10 s, 55°C for 30 s, and 72°C for 30 s, and 72°C for 5 min. After enrichment PCR, 2.5 µl of each library was pooled into a single 1.5 mL tube and purified using the Axygen AxyPrep mag PCR beads with a bead:sample ratio of 1:1. The pooled library was diluted to 3 nM and sequenced on a single lane of an Illumina HiSeq 3000 with single-end (50 bp) reads.

## Multi-cell recordings

Simultaneous whole-cell in vitro recordings were obtained from cortical neurons as previously described (Jiang et al., 2015). Briefly, patch recording pipettes (5–7 MΩ) were filled with internal solution containing 120 mM potassium gluconate, 10 mM HEPES, 4 mM KCl, 4 mM MgATP, 0.3 mM Na$_3$GTP, 10 mM sodium phosphocreatine, Alexa-488 (10 µM) and 0.5% biocytin (pH 7.25). Whole-cell recordings were made from up to eight neurons simultaneously using two Quadro EPC 10 amplifiers (HEKA Electronic, Lambrecht, Germany). A built-in LIH 8+8 interface board (HEKA) was used to achieve simultaneous A/D and D/A conversion of current, voltage, command and triggering signal for up to eight amplifiers. Micromanipulators (Luigs and Neumann) were mounted on a ring specifically designed for multi-patching. PatchMaster software and custom-written Matlab-based programs were used to operate the Quadro EPC 10 amplifiers and perform online and offline analysis of the

data. In order to reveal passive membrane properties and firing patterns of each recorded cell, neurons were stimulated with 600 ms long current pulses starting from −100 to −200 pA with 20 pA steps.

Recordings were made in cortical layers 2/3, 4 and 5, targeting fluorescently labeled (red) cells as well as nearby unlabeled neurons that had clear pyramidal somata and apical dendrites, with the exception of neurons in L4. We visually confirmed successful targeting of tdTomato-expressing neurons based on the spatial overlap of green (due to Alexa-488 in the patch pipette) and red fluorescence (see *Figure 4B*). Unitary excitatory postsynaptic currents (uEPSCs) were evoked by current injection into the presynaptic neurons at 2–3 nA for 2 ms while clamping or holding the membrane potential of the postsynaptic cells at −70 mV. Each neuron was assigned to a laminar position using layer boundaries visible in the high-contrast micrographs obtained during electrophysiological experiments and confirmed post-hoc using the recovered morphology (see below). Latency was defined as the time from the peak of the presynaptic action potential (AP) to 5% of the maximum amplitude of the uEPSC. Amplitude was defined as the maximum amplitude of the uEPSC from baseline. Latency and amplitude are reported as mean ± SD across all connections analyzed.

## Morphological reconstruction after whole-cell multipatch recordings

Light microscopic examination of the morphology and laminar position of each neuron was carried out following previously described protocols (*Jiang et al., 2015*; *Cadwell et al., 2017*). In brief, after in vitro recordings, the slices were fixed by immersion in 2.5% glutaraldehyde/4% paraformaldehyde in 0.1 M phosphate buffer at 4°C for at least 48 hr, and then processed with an avidin-biotin-peroxidase method to reveal cell morphology. The morphologically recovered cells were examined using a 100 × oil immersion objective lens and camera lucida system (Neurolucida, MBF Bioscience). The 3D coordinates and laminar positions of the cells were measured and the distance between each pair of simultaneously recorded neurons was computed, including Euclidean distance, tangential distance (parallel to the pial surface) and vertical distance (perpendicular to the pial surface).

## Quantification of clone size, width, layer restriction, number and type of progenitors

To quantify the width and number of neurons per clone, six near-completely imaged brains were analyzed at P10 using custom-written Matlab software and manual cell segmentation (two brains each treated at E9.5, E10.5 or E11.5, with continuous sections spanning 3–4 mm along the rostro-caudal axis) as follows: a two-dimensional maximum projection of each slice was divided into small sections and presented one at a time to a blinded observer for manual segmentation of neurons throughout the cortex. Glia were excluded based on morphology. Clones were reconstructed across slices by aligning fiducial anatomic landmarks. The number of neurons within each clone was calculated by adding together all of the neurons within the clone across all contiguous slices where the clone was identifiable. On each slice, the widest part of the clone was measured, and the overall width for each clone was computed as the median of the measured width of the clone across all slices. Clone width and number of neurons per clone are reported as the median and interquartile range (IQR, computed using the `median()` and `quantile()` functions in Matlab) in the text and all of the individual data points are shown in *Figure 1D,E*. The number of clones and animals for each treatment condition are reported in the figure legend. The Wilcoxon rank sum test was used to compare E9.5 to E10.5 and E10.5 to E11.5 and those *p*-values are also shown in *Figure 1D,E*.

Clones at P10 were classified as translaminar or layer-restricted after reconstruction by a blinded observer based on whether they spanned all cortical layers (except L1), including L5 and L6 after examination of all consecutive slices containing cells from each clone as well as several flanking slices. The majority of clones (7/11) that were considered layer-restricted are shown in either *Figure 1C* or *Figure 1—figure supplement 1A*. The fraction of all clones that were considered layer-restricted for each treatment condition is reported in *Figure 1F*, as well as the 95% Clopper-Pearson confidence intervals for each ratio. The number of clones and number of animals for each treatment condition are reported in the figure legend. Fisher's exact test was used to compare E9.5 to E10.5 and E10.5 to E11.5 and those *p*-values are also shown in *Figure 1F*.

Clones at E12.5 were examined to determine the number and type of progenitors present in each clone (*Figure 1G–J* and *Figure 1—figure supplement 2*). Radial glial cells were identified as

having their cell body located in the ventricular zone or subventricular zone, with positive nuclear expression of Pax6 but not Tbr2. Intermediate progenitors were identified by the presence of their cell body in either the subventricular zone or intermediate zone and positive nuclear expression of both Pax6 and Tbr2. The entire z-stack was examined, and overlap with nuclear Hoechst staining of tdTomato-positive cells was confirmed for each cell. Clones with ambiguous staining were excluded from the analysis. The number of proliferative units containing tdTomato-positive neurons was also estimated after examination of the entire three-dimensional clone. Although there are no well-established criteria for identifying proliferative units in the developing mouse ventricular zone, if two cells within the same clone were not oriented in the same linear vertical track, we considered them as likely residing in different proliferative units. Examples of clones containing 1–4 radial glial cells, the full range in our dataset, are shown in *Figure 1G,H* and *Figure 1—figure supplement 2*. The distributions of the number of radial glial cells and the number of proliferative units across clones are shown in *Figure 1I,J*. The total number of clones and animals examined are reported in the figure legends.

## Single-cell RNA-sequencing analysis

### Quality control and data pre-processing

From the point of sample collection until the end of the single-cell transcriptomic data pre-processing, all steps were blinded to cell identity with labeled and unlabeled cells intermixed. A total of 278 neurons from 16 translaminar clones were aspirated for single-cell RNA-sequencing experiments. The quality of the full-length cDNA for each sample was analyzed by running on an Agilent bioanalyzer with a High Sensitivity DNA chip. Samples containing less than ~1 ng total cDNA (less than ~67 pg/µl) or with an average size less than 1,500 bp when integrating over the range from 300 to 9,000 bp were not sequenced (~21%, 58/278 neurons, leaving 220 samples).

The final pooled sequencing library was also analyzed on an Agilent Bioanalyzer to confirm that the average library size was less than ~500 bp and there were minimal primer dimers. Reads were aligned to the mouse genome (mm10 assembly) using STAR (v2.4.2a) with default settings. Read counts, rather than RPKMs, were used for the data analysis presented here. Eleven cells were excluded after sequencing due to poor quality sequencing results (~5%, 11/220 neurons, leaving 209 samples; poor quality was defined as greater than three median absolute differences below the median for either total number of reads or total number of genes detected; *Figure 2—figure supplement 1A,B*). Three additional neurons (~1.4%, 3/209) were excluded from further analysis because they had fast-spiking or regular-spiking firing patterns consistent with inhibitory interneurons, leaving 206 samples for all subsequent analyses.

Genes with less than one read per cell on average (*Figure 2—figure supplement 1C*) were removed (n = 12,841 genes remaining) and the count data were normalized using the scran package in R Bioconductor (*Lun et al., 2016*). Quality control plots (*Figure 2C–E*, *Figure 2—figure supplement 1* and *Figure 2—figure supplement 2*) were performed using scran as described in *Lun et al. (2016)*. Across genes, there was a strong correlation between the average count per cell and the number of cells expressing each gene (*Figure 2—figure supplement 1D*) and alternatively filtering genes based on the number of cells expressing each gene had no significant effect on our results, including the GLM prediction of regional identity of neurons in L2/3 and L5 (data not shown). The normalized read counts were used for all subsequent analyses.

### Dimensionality reduction within our dataset

To reduce the dimensionality for visualizing gene expression within our own dataset (*Figure 2F* and *Figure 2—figure supplement 2C*), we used the R Bioconductor implementation of t-distributed Stochastic Neighbor Embedding (t-SNE, `runTSNE()` function of the `scran` package) with the random seed set to 30 for reproducibility. As input, we used the normalized and $\log_2$-transformed counts ('logcounts', *Figure 2—source data 1*) of the top highly variable genes selected with a false discovery rate set to 0.05 (computed using the `correlatePairs()` function with per.gene = TRUE) among the cells being plotted (n = 91 genes). The parameter for perplexity was set to 30. Very similar two-dimensional projections were generated when different parameters or number of genes were used.

## Generalized linear models (GLMs) to predict regional position

We used the `cv.glmnet()` function in R Bioconductor to train a GLM to predict cortical region (*Figure 2G,H*) as follows:

```
cvfit<-cv.glmnet(logcounts,factor,family="multinomial",parallel = TRUE,type.
measure="class",nfolds = 20)
```

The model performance was estimated from the lowest prediction error across all lambda values as follows:

```
perc_correct <- 1-cvfit$cvm[which(cvfit$lambda == cvfit$lambda.min)]
```

To generate a null distribution for each model, we randomly shuffled the cortical region by resampling without replacement 1000 times. For each iteration, the model performance was evaluated as described above. The *p*-values are computed as the fraction of resamples with model performance (percent correct) greater than or equal to the unshuffled model performance. The values in panels 2G and 2H are the unshuffled model performance (in black) and the mean and 95% coverage interval of the shuffled model performances (in gray).

## Mapping to the reference dataset using t-SNE

Using the count matrix of *Tasic et al. (2018)* ($n$ = 23,822, $d$ = 45,768), we selected 3000 'most variable' genes as described in *Kobak and Berens (2019)*. Briefly, we found genes that had, at the same time, high non-zero expression and high probability of near-zero expression. In particular, we excluded all genes that had counts of at least 32 in fewer than 10 cells. For each remaining gene, we computed the mean $\log_2$ count across all counts that were larger than 32 (non-zero expression, $\mu$) and the fraction of counts that were smaller than 32 (probability of near-zero expression, $\tau$). Across genes, there was a clear inverse relationship between $\mu$ and $\tau$, that roughly followed an exponential law $\tau \approx \exp(-1.5 \cdot \mu + a)$ for some horizontal offset $a$. Using a binary search, we found a value $b$ of this offset that yielded 3000 genes with $\tau > \exp(-1.5 \cdot \mu + b) + 0.02$. These 3000 genes were selected as input for dimensionality reduction.

The t-SNE visualizations of the *Tasic et al. (2018)* dataset shown in *Figure 3A*, *Figure 3—figure supplement 1A*, and *Figure 3—figure supplement 2* were generated as described in our previous work (*Kobak and Berens, 2019*). It was computed there using PCA initialization and perplexity combination of 50 and 500, following preprocessing steps of library size normalization (by converting counts to counts per million), feature selection (using the 3000 most variable genes), $\log_2(x+1)$ transformation, and reducing the dimensionality to 50 using PCA. The resulting t-SNE coordinates for all Tasic et al. cells are given in *Figure 3—source data 1*.

Out of 3000 most variables genes selected in the Tasic et al. data set, 1458 genes were present among the 12,841 that we selected in our own data set. We used this set of 1458 genes for the mapping of our cells to the reference data. For each of the $n$ = 206 Patch-seq cells in our dataset, we computed its Pearson correlation with each of the 23,822 reference cells across the 1458 genes, after all counts were $\log_2(x+1)$ transformed. We identified the 10 reference cells with the maximal correlation (10 nearest neighbors of our cell) and positioned our cell at the median t-SNE location of those 10 reference cells (*Kobak and Berens, 2019*).

We performed bootstrapping over genes to estimate the uncertainty of this mapping (*Kobak and Berens, 2019*). Specifically, we selected a bootstrap sample of 1458 genes and repeated the mapping as described above. This was repeated 100 times, to obtain 100 bootstrap positions of each cell. We computed the Euclidean distance between the original mapping position and each of the bootstrap positions and took the 80th percentile of the resulting distribution as a measure of mapping precision. If all bootstrap positions are close to each other, the 80th percentile distance will be small (high precision). If they are far from each other, it will be large (low precision). This measure was used in *Figure 3A* and *Figure 3—figure supplement 1A* to determine the size of each dot representing a mapped cell (values above 10 were plotted as small dots, values greater than five but less than or equal to 10 were plotted as intermediate size dots, and values less than or equal to five were plotted as large dots).

## Mapping to the reference clusters

To assign each of our Patch-seq cells to one of the reference clusters, we log-transformed all counts from *Tasic et al. (2018)* with $\log_2(x+1)$ transformation and averaged the log-transformed counts across all cells in each of the 133 clusters to obtain reference transcriptomic profiles of each cluster, using the same 1458 genes as above (133 × 1458 matrix). We applied the same $\log_2(x+1)$ transformation to the read counts of our Patch-seq cells, and for each cell computed Pearson correlation across the 1458 genes with all 133 *Tasic et al. (2018)* clusters. Each cell was assigned to the cluster to which it had the highest correlation (nearest centroid classifier).

## Probability of related and unrelated neurons mapping to the same clusters

To compute the probability related and unrelated pairs of neurons mapping to the same clusters (*Figure 3C,D* and *Figure 3—figure supplement 1C,D*), we computed the number of pairs mapping to the same cluster as a fraction of all of the pairs analyzed. For *Figure 3C,D* we kept all 133 original clusters. For *Figure 3—figure supplement 1C,D*, we first grouped the 133 transcriptomic clusters into ten broad classes, as labeled in *Figure 3A*. In *Figure 3C* and *Figure 3—figure supplement 1C*, we included all pairs of neurons, whereas in *Figure 3D* and *Figure 3—figure supplement 2D* we included only pairs of neurons in which both cells were positioned within the same cortical layer. The values shown are the overall fraction of pairs mapping to the same cluster or broad class, and the 95% Clopper-Pearson confidence intervals. The *p*-values are computed using the Chi-squared test.

Since we did not know a priori what the baseline distribution of cell types for unrelated neurons would be, we could not prospectively estimate the optimal sample size for detecting a difference between related and unrelated neurons in mapping to the same cell type. However, after collecting the data and finding that the probability of unrelated neurons matching to the same cell type is approximately 8% (*Figure 3C*), we can compute the sample size needed to detect a 10% increase in the probability of matching to the same cell type (a conservative estimate of meaningful effect size) with alpha = 0.05% and 90% statistical power as 233. Given that our sample sizes are 337 and 409 for related and unrelated neurons, respectively, we are appropriately powered to detect a 10% difference in the probability of related neurons to map to the same transcriptomic cell type.

## Comparison of connection probability between related and unrelated neurons

### Comparison using raw data

Related pairs were defined as pairs in which both the pre- and post-synaptic neurons were tdTomato-positive excitatory neurons organized in a well-isolated radial unit of labeled cells (>300 µm separation from other labeled clones). Control pairs were defined as pairs of nearby excitatory neurons in which one cell was tdTomato-positive (either the pre- or post-synaptic cell, but not both) and one was tdTomato-negative. The connection probability was determined as the total number of connections divided by the total number of connections tested within each category (all connections tested, only vertical connections, only lateral connections, or individual layer-defined connection types). The values shown in *Figures 4D–F* and *5C,D* are the connection probability and 95% Clopper-Pearson confidence intervals. The number of connections tested for each category is reported in the figure legends. Fisher's exact test was used to compare the connection probabilities between related and unrelated cells and those *p*-values are shown in *Figures 4D–F* and *5C,D*.

### Power analysis

For our connectivity experiments, we predicted based on a prior study that there would be overall a ~ 5.7 fold increase in connectivity for clonally related neurons (from 6.3% to 36.3%; *Yu et al., 2009*). The minimal sample size to detect a difference of this magnitude with 90% statistical power and alpha = 0.05 is 34 in each group. Each group in our study (related and unrelated, for each layer-connection type, a 2 × 9 matrix) exceeds this sample size, often by two- or three-fold. Thus, we are appropriately powered to capture this previously reported effect size for increased connectivity between clonally related neurons for each layer-specific connection examined.

One possible confound to this estimation could arise if the previously reported connection probabilities only apply to related neurons derived from the same radial glial cell (RGC). In that case, since our clones contain one average two radial glial progenitors, the effect of lineage on connectivity

may be diluted, and such an effect could more severely impact lateral connections compared to vertical connections. To address this concern, we also estimated our statistical power to detect a potential change in lateral connectivity between RGC-related and unrelated neurons (*Figure 5—figure supplement 1*), and computed the predicted lateral connection probability, $P_{rl}$ using the following equation:

$$\mathrm{P_{rl}} = ((\mathrm{P_{ul}} \cdot \mathrm{S/L} \cdot (\mathrm{N} - 1)/\mathrm{N}) + (\mathrm{P_{ul}} \cdot \mathrm{FC} \cdot (\mathrm{S/(NL)} - 1)))/(\mathrm{S/L} - 1),$$

where $P_{ul}$ is the measured lateral connection probability for unrelated neurons (51/626), *S* is the clone size (set to our median clone size of 60), *L* is the number of layers (set to 4), *N* is the number of radial glial lineages within each clone, and *FC* is the estimated fold change in connectivity between RGC-related and unrelated neurons. We subtract one in the second term in the nominator and in the denominator to account for the fact that each neuron can only laterally connect to *other* neurons derived from the same RGC, i.e. it is itself excluded from the pool of *S/(NL)* RGC-related neurons in the same layer. For simplicity, this equation assumes an equal distribution of RGC-related neurons across layers, but the results would not change if layer-restriction of individual RGC lineages were introduced (data not shown).

For each possible combination of *N* ranging from 1 to 20 and *FC* ranging from 0 to 30, we used the above equation to estimate the predicted $P_{rl}$ and calculated the effect size, *H*, using Cohen's h values, as follows:

$$\mathrm{H} = 2 \cdot \arcsin(\mathrm{sqrt}(\mathrm{P_{ul}})) - 2 \cdot \arcsin(\mathrm{sqrt}(\mathrm{P_{rl}})).$$

The statistical power was then computed using the `pwr.2p2n.test()` function in R Bioconductor with `h` set to *H*, `n1` set to 626 (the number of unrelated lateral connections tested), `n2` set to 248 (the number of related lateral connections tested), and `sig.level` set to 0.05. The resulting $P_{rl}$ and statistical power values are plotted in *Figure 5—figure supplement 1A,B* as a function of fold change (*FC*) with a separate curve for each value of *N*.

We also computed the value of *FC* that would explain our observed vertical connectivity rates in the setting of pooling multiple RGC lineages using the following equation:

$$\mathrm{P_{rv}} = \mathrm{P_{uv}} \cdot (\mathrm{N} - 1)/\mathrm{N} + \mathrm{FC} \cdot \mathrm{P_{uv}} \cdot 1/\mathrm{N},$$

where $P_{rv}$ is the measured vertical connection probability for all labeled neurons (28/464) and $P_{uv}$ is the measured vertical connection probability for unrelated neurons (19/711). Using this equation, one can compute a value of *FC* for each possible value of *N*. This value can then be used to infer the value of $P_{rl}$ that would be expected if the increase in connectivity among RGC-related neurons is the same for vertical and lateral connections, using the same equation as above. In *Figure 5—figure supplement 1* the red dots show, for each value of *N*, the values of $P_{rl}$ and *FC* that would be expected based on the observed vertical connection probabilities. The blue dashed lines correspond to our observed values of $P_{rl}$ and *FC* for lateral connections (assuming *N* = 1 for estimating *FC*).

## Comparison to distance-matched controls

Related pairs were defined as above. In contrast to the above comparison, control pairs were defined as pairs of excitatory neurons in which one *or both* cells were tdTomato-negative, to increase the number of available controls for distance-matching (a caveat is that two tdTomato-negative cells can in principle belong to another, unlabeled clone, but we consider this to be unlikely). For each related pair, we identified a set of 'matched' control pairs for which the pre- and post-synaptic neurons were located in the same cortical layers as the pre- and post-synaptic neurons of the related pair, and for which both the tangential and vertical distances between the control cells were within 20 μm of the analogous distances between the two related cells. Related pairs that did not have any matching control pairs fitting these criteria were excluded from further analysis.

To compare connectivity between related and distance-matched control pairs, we used bootstrapping over related pairs. Specifically, on each of the 1000 iterations, we drew a bootstrap sample (resample with replacement) from the set of related pairs, and selected one matched control pair for each related pair that was selected. Values in *Figure 5—figure supplement 2* are the mean and 95% coverage intervals across resamples of related and matched control connection probabilities. For each resample, we also computed the difference between the related and matched control

connection probabilities. We 'inverted' the bootstrap confidence interval for this difference to estimate the *p*-value. Specifically, the mean true difference in connection probability was first subtracted from all bootstrapped differences, and the *p*-value was estimated as the fraction of resampled differences with absolute value greater than or equal to the true mean difference (two-tailed test). These *p*-values are shown in *Figure 5—figure supplement 2*.

## Comparison at different rostrocaudal positions

To determine whether the effect of cell lineage varied according to rostrocaudal position, clones were sorted into two groups based on their rostrocaudal position ('rostral' includes clones within SS1 proper but also other rostral cortical areas, and similarly for 'caudal' clones and V1). The values shown in *Figure 5—figure supplement 3* are the connection probability and 95% Clopper-Pearson confidence intervals for each group. The number of connections tested in each category is reported in the figure legend. Fisher's exact test was used to compare the connection probabilities between related and unrelated cells in each group and those *p*-values are shown in *Figure 5—figure supplement 3*.

## Generalized linear model of connection probability

We also built a generalized linear model (GLM) to explain connection probability (*P*) as a function of connection class, lineage relationship, Euclidean distance between the cells, and rostrocaudal position (a numeric value ranging from 1 to 5 with 1 being most rostral and 5 being most caudal). We fit a binomial GLM (using the `glmfit()` function in Matlab) containing the relevant linear terms and all possible pairwise interactions:

$$g(P) = \beta_0 + \beta_L \cdot L + \beta_C \cdot C + \beta_D \cdot D + \beta_R \cdot R + \beta_{LC} \cdot L \cdot C + \beta_{LD} \cdot L \cdot D$$
$$+ \beta_{LR} \cdot L \cdot R + \beta_{CD} \cdot C \cdot D + \beta_{CR} \cdot C \cdot R + \beta_{DR} \cdot D \cdot R$$

where $\beta_0$ is a constant term, *L* is a binary variable representing the lineage relationship (1 for related and 0 for unrelated), *C* is a binary variable representing the connection type (1 for vertical and 0 for lateral), *D* is the Euclidean distance between the cells in microns, *R* is a numeric variable representing the rostrocaudal position of the clone with integer values from 1 (most rostral) to 5 (most caudal), and $\beta_i$ are the corresponding coefficients. The presence of a connection was modeled as Bernoulli distributed with probability *P*, using the logit link function, g(*P*)=ln(*P*/(1 *P*)). The estimated coefficients and *p*-values of each term are reported in *Table 1*.

## Simple connectivity model to estimate the expected input from related cells

For a particular postsynaptic cell in layer $j \in \{\mathrm{L2/3, L4, L5}\}$, we modeled the number of input connections from cells in a particular layer *i* and a particular lineage relation $l \in \{\mathrm{related, unrelated}\}$ as a binomial distribution $\mathrm{B}(n_{il}, p_{il})$. The probabilities $p_{ijl}$ were set to the measured connection probabilities. The pool sizes $n_{il}$ were set to the product $n_{il} = n_i q_l$ of the number of cells $n_i$ residing in the particular input layer and the fraction $q_l$ of cells with that particular lineage. To compute $n_i$, we assumed that a cortical slab of 1mm$^2$ contains about 100,000 neurons and that 80% of those are excitatory neurons. We further assumed that a particular cell only connects to other cells within a tangential radius of $\mathrm{r}$, which we set to half the 99% quantile of pairwise distances measured in our dataset (r = 0.087mm). The resulting cylinder of cortex contained $\pi r^2 \times 80,000 \approx 1,908$ excitatory neurons. We assumed that 35% of these cells reside in L2/3, 15% in L4, 25% in L5, and 25% in L6. The fraction $q_c$ of related cells in that cylinder was computed as the ratio $q_r = \frac{k}{N}$ of the median clone size ($k = 60$) and the number of cells in the cortical cylinder (N = 1908). All model computations were performed using Python.

We computed the expected fraction of related cells in the input connections to a particular cell (*Figure 4G* and *Figure 5E*) as,

$$e_{ij} = \frac{p_{ijr} n_{ir}}{p_{ijr} n_{ir} + p_{iju} n_{iu}} = \frac{p_{ijr} q_r}{p_{ijr} q_r + p_{iju}(1 - q_r)},$$

where subscript *r* refers to related neurons and *u* to unrelated neurons. Note that $q_u = 1 - q_r$. We

propagated the standard error from $p_{ijl}$ to $e_{ij}$ using a first order Taylor approximation: In general, the propagated variance of a function $f(X,Y)$ of two random variables is given by

$$\mathrm{Var}[F] \approx F_x^2 \sigma_x^2 + F_y^2 \sigma_y^2 + 2 F_x F_y \sigma_{xy}$$

where $F_x$ and $F_y$ denote the partial derivatives of $\mathrm{F}$ with respect to the variables in the subscript, and $\sigma_x^2$, $\sigma_y^2$, and $\sigma_{xy}$ denote the variances and covariance of $X$ and $Y$ (*Lee and Forthofer, 2006*). In our case, the random variables are the estimators $\hat{p}_{ijl}$ of the connection probabilities which have a variance (squared standard error) of $\sigma_{ijl}^2 = \frac{\hat{p}_{ijl}(1-\hat{p}_{ijl})}{m_{ijl}}$ and no covariance because we assume that the two different lineages were measured independently. The denominator $m_{ijl}$ denotes the number of tested connections for that particular lineage and combination of layers. This yields the following standard error for $e_{ij}$.

$$\mathrm{SE}[e_{ij}] = \frac{q_c^2(1-q_c)^2\left(\hat{p}_{iju}^2\frac{\hat{p}_{ijc}(1-\hat{p}_{ijc})}{n_{ic}} + \hat{p}_{ijc}^2\frac{\hat{p}_{iju}(1-\hat{p}_{iju})}{n_{iu}}\right)}{\left(\hat{p}_{iju}(1-q_c) + \hat{p}_{ijc}q_c\right)^4}$$

The values reported in *Figure 4G* and *Figure 5E* are the estimates and propagated standard errors.

### Log-ratios of connection probabilities between related and unrelated neurons

To visualize the overall pattern of the differences in connectivity between related and unrelated neurons, we generated a heatmap of the log ratio of connection probabilities for related and unrelated neurons (*Figure 5F*). For each layer-defined connection type, we took the $\log_2$ of the ratio of the related pair connection probability and unrelated pair connection probability, with Laplace smoothing (by adding 1 to both the numerator and denominator). Specifically, if $A$ out of $B$ related pairs and $C$ out of $D$ unrelated pairs were connected, we computed the log-ratio as $\log_2\{[(A+1)/(B+1)] / [(C+1)/(D+1)]\}$. The 95% confidence intervals were computed via bootstrapping. For each bootstrap iteration, we generated $A_{\mathrm{boot}}$ as a binomial draw with $p=(A+1)/(B+1)$ and $n = B$, and $C_{\mathrm{boot}}$ as a binomial draw with $p=(C+1)/(D+1)$ and $n = D$. As the 95% confidence interval, we took 95% coverage interval of the bootstrapped log-ratios $\log_2\{[(A_{\mathrm{boot}}+1)/(B+1)] / [(C_{\mathrm{boot}}+1)/(D+1)]\}$.

### Data and software availability

The single-cell RNA-seq data is deposited in GEO under accession code GSE140946. Our custom-written software used for analyses and for generating figures is available on Github at https://github.com/atlab/commons and https://github.com/crcadwell/cadwell2020 (*Cadwell and Sinz, 2020*; copy archived at https://github.com/elifesciences-publications/cadwell2020).

## Acknowledgements

We thank Alexander Ecker, Jacob Reimer, Dimitri Yatsenko, Shan Shen, Emmanouil Froudarakis, Amy Morgan, Camila Lopez, Megan Rech, and Shannon McDonnell for helpful discussions and technical support. We also thank Tomasz Nowakowski for critical reading of the manuscript and for sharing equipment.

This project was supported by the Optical Imaging and Vital Microscopy core at Baylor College of Medicine; grants R01MH109556, R01MH103108, R01DA028525, DP1EY023176, P30EY002520, T32EY07001, and DP1OD008301 from the National Institutes of Health (NIH) to AST; grant 707359 from the National Science Foundation to AST; grants from the Swedish Research Council and the Vallee Foundation to RS; grants from the Deutsche Forschungsgemeinschaft (DFG, EXC 2064, BE5601/4-1) and the German Federal Ministry of Education and Research (BMBF; FKZ 01GQ1601) to PB; the McKnight Scholar Award to AST; and the Arnold and Mabel Beckman Foundation Young Investigator Award to AST X J was supported by BCM Faculty Start-up Fund. CRC was supported by NIH grants F30MH095440, T32GM007330 and T32EB006350. PGF was supported by NIH grant F30MH112312. CRC and PGF were both supported by the Baylor Research Advocates for Student Scientists (BRASS) foundation. FHS is funded by the Institutional Strategy of the University of

Tübingen (Deutsche Forschungsgemeinschaft, ZUK 63) and the Carl-Zeiss-Stiftung. FHS acknowledges support from the DFG Cluster of Excellence ''Machine Learning – New Perspectives for Science'' EXC 2064/1, project number 390727645, and Amazon through an AWS Machine Learning Research Award.

This project was also supported by the Intelligence Advanced Research Projects Activity (IARPA) via Department of Interior/Interior Business Center (DoI/IBC) contract number D16PC00003. The U. S. Government is authorized to reproduce and distribute reprints for Governmental purposes notwithstanding any copyright annotation thereon. Disclaimer: The views and conclusions contained herein are those of the authors and should not be interpreted as necessarily representing the official policies or endorsements, either expressed or implied, of IARPA, DoI/IBC, or the U.S. Government.

This project was also supported by the National Institute of Mental Health and the National Institute of Neurological Disorders and Stroke under Award Number U19MH114830. The content is solely the responsibility of the authors and does not necessarily represent the official views of the National Institutes of Health.

## Additional information

### Funding

| Funder | Grant reference number | Author |
| --- | --- | --- |
| Baylor College of Medicine | Optical Imaging and Vital Microscopy Core | Andreas Savas Tolias |
| National Institutes of Health | R01MH103108 | Andreas Savas Tolias |
| National Institutes of Health | R01DA028525 | Andreas Savas Tolias |
| National Institutes of Health | DP1EY023176 | Andreas Savas Tolias |
| National Institutes of Health | P30EY002520 | Andreas Savas Tolias |
| National Institutes of Health | T32EY07001 | Andreas Savas Tolias |
| National Institutes of Health | DP1OD008301 | Andreas Savas Tolias |
| National Science Foundation | 707359 | Andreas Savas Tolias |
| Svenska Forskningsrådet Formas | | Rickard Sandberg |
| Vallee Foundation | | Rickard Sandberg |
| Deutsche Forschungsgemeinschaft | EXC 2064 | Philipp Berens |
| Deutsche Forschungsgemeinschaft | BE5601/4-1 | Philipp Berens |
| Bundesministerium für Bildung und Forschung | FKZ 01GQ1601 | Philipp Berens |
| McKnight Foundation | McKnight Scholar Award | Andreas Savas Tolias |
| Arnold and Mabel Beckman Foundation | Young Investigator Award | Andreas Savas Tolias |
| Baylor College of Medicine | Faculty start-up fund | Xialong Jiang |
| National Institutes of Health | F30MH095440 | Cathryn R Cadwell |
| National Institutes of Health | T32GM007330 | Cathryn R Cadwell |
| National Institutes of Health | F30MH112312 | Paul G Fahey |
| Baylor College of Medicine | BRASS Scholar Award | Cathryn R Cadwell Paul G Fahey |
| Deutsche Forschungsgemeinschaft | ZUK 63 | Fabian H Sinz |
| Carl Zeiss Stiftung | | Fabian H Sinz |
| National Institutes of Health | R01MH109556 | Andreas Savas Tolias |

The funders had no role in study design, data collection and interpretation, or the decision to submit the work for publication.

## Author contributions

Cathryn R Cadwell, Conceptualization, Data curation, Software, Formal analysis, Funding acquisition, Investigation, Methodology; Federico Scala, Data curation, Investigation, Methodology; Paul G Fahey, Data curation, Investigation, Methodology, Writing - review and editing; Dmitry Kobak, Data curation, Formal analysis, Visualization, Writing - review and editing; Shalaka Mulherkar, Data curation, Investigation, Writing - review and editing; Fabian H Sinz, Formal analysis, Supervision, Writing - review and editing; Stelios Papadopoulos, Zheng H Tan, Data curation, Validation, Investigation; Per Johnsson, Resources, Data curation, Investigation, Methodology; Leonard Hartmanis, Resources, Data curation, Investigation; Shuang Li, Data curation, Investigation; Ronald J Cotton, Conceptualization, Guidance on analyses; Kimberley F Tolias, Conceptualization, Resources, Supervision, Writing - review and editing; Rickard Sandberg, Conceptualization, Resources, Data curation, Formal analysis, Supervision, Funding acquisition, Investigation, Methodology, Writing - review and editing; Philipp Berens, Conceptualization, Resources, Data curation, Formal analysis, Supervision, Funding acquisition, Investigation, Writing - review and editing; Xiaolong Jiang, Conceptualization, Resources, Data curation, Formal analysis, Supervision, Investigation, Methodology, Writing - review and editing; Andreas Savas Tolias, Conceptualization, Resources, Data curation, Software, Formal analysis, Supervision, Funding acquisition, Investigation, Methodology, Writing - review and editing

## Author ORCIDs

Cathryn R Cadwell https://orcid.org/0000-0003-1963-8285
Dmitry Kobak https://orcid.org/0000-0002-5639-7209
Shalaka Mulherkar http://orcid.org/0000-0001-8736-527X
Fabian H Sinz https://orcid.org/0000-0002-1348-9736
Kimberley F Tolias http://orcid.org/0000-0002-2092-920X
Philipp Berens http://orcid.org/0000-0002-0199-4727
Andreas Savas Tolias https://orcid.org/0000-0002-4305-6376

## Ethics

Animal experimentation: This study was performed in strict accordance with the recommendations in the Guide for the Care and Use of Laboratory Animals of the National Institutes of Health. All of the animals were handled according to an approved institutional animal care and use committee (IACUC) protocol of Baylor College of Medicine (protocol # AN-4703). Every effort was made to minimize suffering.

## Decision letter and Author response

Decision letter https://doi.org/10.7554/eLife.52951.sa1
Author response https://doi.org/10.7554/eLife.52951.sa2

# Additional files

## Supplementary files

• Transparent reporting form

## Data availability

Sequencing data have been deposited in GEO under accession code GSE140946. All data generated or analyzed during this study are included in the manuscript and supporting files. Source data files have been provided for Figures 1, 2, 3 and 4. The source data provided for Figure 4 also apply to Figure 5 and Table 1.

The following dataset was generated:

| Author(s) | Year | Dataset title | Dataset URL | Database and Identifier |
|---|---|---|---|---|
| Cadwell CR, Scala F, Fahey PG, Kobak D, Mulherkar S, Sinz FH, Papadopoulos S, Tan ZH, Johnsson P, Hartmanis L, Li S, Cotton RJ, Tolias KF, Sandberg R, Berens P, Jiang X, Tolias AS | 2019 | Cell type composition and circuit organization of neocortical radial clones | http://www.ncbi.nlm.nih.gov/geo/query/acc.cgi?acc=GSE140946 | NCBI Gene Expression Omnibus, GSE140946 |

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
