## [Decision Letter]

**Acceptance summary:**

This study examines the relationship of clonality with connectivity in the mammalian cortex. The authors combine genetic labeling of neurons with patch recordings and transcriptomic sequencing to assign layer identity. Their evidence that co-labeled neurons are more likely to be connected across rather than within layers clarifies the understanding of cortical organization during development.

**Decision letter after peer review:**

Thank you for submitting your article "Cell type composition and circuit organization of clonally related excitatory neurons in the juvenile mouse neocortex" for consideration by *eLife*. Your article has been reviewed by three peer reviewers, one of whom is a member of our Board of Reviewing Editors, and the evaluation has been overseen by Timothy Behrens as the Senior Editor. The following individuals involved in review of your submission have agreed to reveal their identity: Robert Hevner (Reviewer #2); Gordon Fishell (Reviewer #3).

The reviewers have discussed the reviews with one another and the Reviewing Editor has drafted this decision to help you prepare a revised submission.

Summary:

In this study, the authors use tamoxifen induction at e10 for lineage tracing of Nestin+ radial glial progenitor-derived clones in the juvenile mouse neocortex. They characterize the size and morphology of Nestin+ clones and, using Patch-Seq combined patch clamping/snRNAseq method, give a limited analysis of excitatory cell type composition within these clones. They employ multi-cell patch clamp recording of clonally related and non-clonally related neurons across each neocortical lamina to show clonally-related neurons are more likely to be connected vertically across layers than horizontally within layers, though connectivity is overall quite low. They also present a general linear model showing an interaction between excitatory neuron lineage and connectivity. The authors findings largely concur with findings in the literature but expand and clarify them in important ways because of the large, high quality dataset they provide. The reviewers had no major concerns with the execution of the experiments but did feel that the authors need to address two points regarding the interpretation of their findings.

Essential revisions:

1) The reviewers questioned the extent to which the clones labeled by the E10.5 tamoxifen strategy arose from symmetric versus asymmetric divisions. This distinction is important because symmetric divisions arise from neural stem cells, thus their progeny have no particular relationship to the post mitotic progeny they give rise to.

One reviewer stated the concern saying, "The fact their clones span on the order of 300 microns suggests to me that they are often grouping cells resulting from more than one radial glia and hence while lineage related are likely not clonal in the sense they arise from a single asymmetric progenitor."

The interpretation of the lineage relationship of the labeled neurons is important for the field because it impacts thinking about the causality of connectivity. The text frequently alludes to the role of lineage in directing siblings to preferentially make connections in the vertical but not horizontal axes. However whether it is lineage per se or the proximity of siblings that results in their increased connectivity is unknowable and a confound in the current literature in this field.

The reviewers suggested that this concern could be addressed experimentally by comparison of clones labeled at E10.5, 11.5 and 12.5 rather than focusing on connectivity of only E10.5 (given that the clones labeled at later times are more certainly arising from single radial glial precursors). If the authors have this data, the reviewers would encourage them to make the direct comparison as this would be an elegant approach to resolving the concern. Alternatively, the authors could address this concern about the relationship of the labeled neurons with text revisions alone by acknowledging they may underestimate the predictive power of clones because of the mix of symmetric and asymmetric divisions.

2) One concern and one related opportunity were raised regarding the transcriptomic data.

When the Patch-Seq data were aligned to a t-SNE plot for adult C57B6/J mouse neocortex atlas from Tasic et al., 2018, there was underwhelming matching of individual layers to the expected reference layers. This could be related to the age of the mice. However, given then comment in the text, "The discrepancies were mostly due to some neurons mapping to a transcriptomic type from a neighboring layer." This leads to the concern: how certain is it that cells were collected from each specific layer and that there is an increase in, say, L4 to L5 connections and not truly between two L4 cells or two L5 cells?

A second reviewer offered a possible approach, saying, "As they have the transcriptomic identity of each of these types, do they indicate that certain transcriptomic types connect with similar types with higher probability? This is something that almost certainly must be true and their study has the potential to identify whether vertical integration reflects connectivity of subtypes, which would be incredibly valuable to understand."

We encourage the authors to analyze the transcriptome classification of the synaptically interconnected cells as a means to resolve the concerns about the identity of neurons within and between the layers and to offer further insights into connectivity. We appreciate that the data may not yield a clear answer and would accept text revisions to acknowledge the possible confounds in assignment of the cell types recorded.

---

## [Author Response]

Essential revisions:1) The reviewers questioned the extent to which the clones labeled by the E10.5 tamoxifen strategy arose from symmetric versus asymmetric divisions. This distinction is important because symmetric divisions arise from neural stem cells, thus their progeny have no particular relationship to the post mitotic progeny they give rise to.One reviewer stated the concern saying, "The fact their clones span on the order of 300 microns suggests to me that they are often grouping cells resulting from more than one radial glia and hence while lineage related are likely not clonal in the sense they arise from a single asymmetric progenitor."The interpretation of the lineage relationship of the labeled neurons is important for the field because it impacts thinking about the causality of connectivity. The text frequently alludes to the role of lineage in directing siblings to preferentially make connections in the vertical but not horizontal axes. However whether it is lineage per se or the proximity of siblings that results in their increased connectivity is unknowable and a confound in the current literature in this field.The reviewers suggested that this concern could be addressed experimentally by comparison of clones labeled at E10.5, 11.5 and 12.5 rather than focusing on connectivity of only E10.5 (given that the clones labeled at later times are more certainly arising from single radial glial precursors). If the authors have this data, the reviewers would encourage them to make the direct comparison as this would be an elegant approach to resolving the concern. Alternatively, the authors could address this concern about the relationship of the labeled neurons with text revisions alone by acknowledging they may underestimate the predictive power of clones because of the mix of symmetric and asymmetric divisions.

The reviewers are correct that many progenitors present at E10.5 are likely neuroepithelial stem cells (NSCs). To directly address this issue of whether clones labeled at E10.5 are derived from radial glial cells (RGCs) or NSCs, we completed a series of experiments in which we sacrificed animals at E12.5 and performed immunohistochemical staining for Pax6 (positive in RGCs and intermediate progenitors) and Tbr2 (positive in intermediate progenitors but not RGCs). We found that our E10.5-treated clones were composed of two radial glial lineages on average (range 1-4), suggesting that our lineage tracing method labeled primarily NSCs in their final cycle of symmetric cell division. This data was already included in our original submission to *eLife* (Figures 1G-J and Figure 1—figure supplement 2), but we now emphasize this finding more clearly in the paper. Therefore, our clones are indeed composed of neurons that are more distantly related to one another (i.e. "cousins" rather than "sisters") compared to prior studies examining connectivity between clonally related excitatory neurons (Yu et al., 2009; He et al., 2015).

The reviewers also raise an important point regarding the implications of our results, and we agree that proximity during neurogenesis and migration may play an important role in promoting the formation of synaptic connections between clonally related excitatory neurons. If RGC lineage were the only factor driving the development of synaptic specificity among related neurons, then we would expect the influence of lineage on connectivity to be more diluted within our more distantly related clones compared to closely related cells derived from the same RGC. Given that connectivity between clonally related excitatory neurons labeled at E12.5 has already been extensively studied (Yu et al., 2009; He et al., 2015), we feel that repeating our connectivity experiments at E11.5 and E12.5 would yield little additional insight and is well beyond the scope of the current manuscript. Instead of repeating these experiments ourselves, we utilized the published data from Yu et al., 2009 to determine whether our results could be explained by a dilution of the effect size reported for a single radial glial lineage, and the answer is clearly negative (presented in our original Figures 4G and 5E). For some connection types (specifically projections from layers 2/3 and 4 to layer 5) there is nearly an order of magnitude difference in the total fraction of input attributed to clonally related cells in our data than would be predicted by simply pooling together multiple RGC-derived clones (Figure 4G). Thus, our results suggest that more distant lineage relationships (i.e. prior to the onset of asymmetric cell divisions) and/or physical proximity of newborn neurons indeed contribute to the formation of lineage-associated preferential connectivity. For these reasons, we believe it is precisely *because* we labeled progenitors so early in development that our results are somewhat surprising and challenge the current dogma.

In summary, we show that our clones labeled at E10.5 consist of more than one radial glial lineage (two on average), and that the connectivity rates we observe cannot be explained by pooling of multiple RGC-derived clones based on previously reported connectivity data obtained at E10.5 (Yu et al., 2009). We agree that dissociation of the influences of cell lineage and physical proximity presents a major challenge to the entire field, and our study highlights the need for further mechanistic studies on this important issue. We have now revised the text to more clearly emphasize the hypothesis that proximity of siblings during neurogenesis and migration, rather than lineage *per se*, could explain our observations. We believe this revised discussion more clearly articulates the significance and implications of our findings and we thank the reviewers for pointing out this issue.

2) One concern and one related opportunity were raised regarding the transcriptomic data.When the Patch-Seq data were aligned to a t-SNE plot for adult C57B6/J mouse neocortex atlas from Tasic et al., 2018, there was underwhelming matching of individual layers to the expected reference layers. This could be related to the age of the mice. However, given then comment in the text, "The discrepancies were mostly due to some neurons mapping to a transcriptomic type from a neighboring layer." This leads to the concern: how certain is it that cells were collected from each specific layer and that there is an increase in, say, L4 to L5 connections and not truly between two L4 cells or two L5 cells?A second reviewer offered a possible approach, saying, "As they have the transcriptomic identity of each of these types, do they indicate that certain transcriptomic types connect with similar types with higher probability? This is something that almost certainly must be true and their study has the potential to identify whether vertical integration reflects connectivity of subtypes, which would be incredibly valuable to understand."We encourage the authors to analyze the transcriptome classification of the synaptically interconnected cells as a means to resolve the concerns about the identity of neurons within and between the layers and to offer further insights into connectivity. We appreciate that the data may not yield a clear answer and would accept text revisions to acknowledge the possible confounds in assignment of the cell types recorded.

The reviewers raise a valid concern. However, we would point out that the layer assignments in the Tasic et al., 2018 dataset are based on layer-enriched microdissections and are not always precise measurements of the location of the cell bodies. Moreover, careful examination of the data presented in Tasic et al. reveals that many transcriptomic excitatory cell types are not perfectly restricted to a single layer, despite the designation of a single layer in their name. This includes several of the most prevalent cell types in our dataset, for example: L4 IT VISp Rspo1 cells were present in L2/3 as well as in L4; L5 IT VISp Hdd11b1 Endou cells were found in nearly equal numbers in L4 and L5; and L6 IT Col18a1 cells were similarly present in both layers 5 and 6 (see Tasic et al., Figure 4B). Thus, our finding that some cells located in one physical layer position mapped to a transcriptomic cell type with a slightly different "layer" designation in the transcriptomic name is not all that surprising and is completely in keeping with the degree of variation in physical location of the cell body described in the reference dataset. We have revised the quoted sentence to reflect that this "mismatch" likely represents a true difference between the physical location of the neuron and the transcriptomic cell type "layer" designation, rather than an error in the assignment of physical layer position.

The physical location assignment of each neuron was determined by an expert electrophysiologist using differential interference contrast microscopy at the time of the experiment and later confirmed using biocytin staining as is the standard in the field (Jiang et al., 2015; Cadwell et al., 2017; Scala et al., 2019). The process is relatively straightforward, but nonetheless, in response to the reviewers’ concerns we have more clearly explained the procedure in the main text and Materials and methods sections. We see no reason to doubt the layer assignment of the cells in our analysis. Moreover, it seems highly unlikely that the increase in connectivity from L4 to L5 that we observed between related neurons could be explained by erroneous layer assignments given that there was no increase in connectivity between related L4 cells or between related L5 cells themselves.

The suggestion to analyze the transcriptome of the connected cells directly would be an ideal experiment, and something we would absolutely love to do. Unfortunately, the reviewers might have misunderstood the nature of our dataset – we do not have connectivity and transcriptome data for the same set of cells, instead we collected connectivity data for some cells and transcriptomic data for a completely different set of cells. To the best of our knowledge, no one has yet been able to combine Patch-seq with multiple whole-cell recordings. Both of these techniques are technically quite challenging by themselves, and the modifications that make one feasible tend to make the other more difficult. Most notably, the changes to the internal solution necessary to obtain high-quality RNA make it more difficult to patch the cells for a long period of time. Conversely, stable and long recordings are the key to success in multi-patching experiments. We and others are actively working to combine the two techniques, but have not yet found a way to perform it routinely. Thus, our current dataset does not allow a direct correlation of the transcriptome with connectivity.